# Pharmacological inhibition of cystine–glutamate exchange induces endoplasmic reticulum stress and ferroptosis

Scott J Dixon[1†‡], Darpan N Patel[1†], Matthew Welsch[1], Rachid Skouta[1], Eric D Lee[1], Miki Hayano[1], Ajit G Thomas[2], Caroline E Gleason[1], Nicholas P Tatonetti[3,4,5], Barbara S Slusher[2,6], Brent R Stockwell[1,5,7,8]*

[1]Department of Biological Sciences, Columbia University, New York, United States; [2]Brain Science Institute, Johns Hopkins Medicine, Baltimore, United States; [3]Department of Biomedical Informatics, Columbia University, New York, United States; [4]Department of Medicine, Columbia University, New York, United States; [5]Department of Systems Biology, Columbia University, New York, United States; [6]Department of Neurology, Johns Hopkins Medicine, Baltimore, United States; [7]Department of Chemistry, Columbia University, New York, United States; [8]Howard Hughes Medical Institute, Columbia University, New York, United States

**Abstract** Exchange of extracellular cystine for intracellular glutamate by the antiporter system $x_c^-$ is implicated in numerous pathologies. Pharmacological agents that inhibit system $x_c^-$ activity with high potency have long been sought, but have remained elusive. In this study, we report that the small molecule erastin is a potent, selective inhibitor of system $x_c^-$. RNA sequencing revealed that inhibition of cystine–glutamate exchange leads to activation of an ER stress response and upregulation of *CHAC1*, providing a pharmacodynamic marker for system $x_c^-$ inhibition. We also found that the clinically approved anti-cancer drug sorafenib, but not other kinase inhibitors, inhibits system $x_c^-$ function and can trigger ER stress and ferroptosis. In an analysis of hospital records and adverse event reports, we found that patients treated with sorafenib exhibited unique metabolic and phenotypic alterations compared to patients treated with other kinase-inhibiting drugs. Finally, using a genetic approach, we identified new genes dramatically upregulated in cells resistant to ferroptosis.

*For correspondence:
bstockwell@columbia.edu

†These authors contributed equally to this work

Present address: ‡Department of Biology, Stanford University, Stanford, United States

## Introduction

Transporters for small molecule nutrients, including sugars, nucleotides, and amino acids, are essential for cellular metabolism and represent potential targets for drug development (*Hediger et al., 2013*). System $x_c^-$ is a cell-surface Na⁺-independent cystine–glutamate antiporter composed of the 12-pass transmembrane transporter protein SLC7A11 (xCT) linked via a disulfide bridge to the single-pass transmembrane regulatory subunit SLC3A2 (4F2hc) (*Sato et al., 1999*; *Conrad and Sato, 2012*). System $x_c^-$ is required for normal mammalian blood plasma redox homeostasis, skin pigmentation, immune system function, and memory formation (*Chintala et al., 2005*; *Sato et al., 2005*; *De Bundel et al., 2011*). Aberrant system $x_c^-$ function is implicated in tumor growth and survival, cancer stem cell maintenance, drug resistance, and neurological dysfunction (*Okuno et al., 2003*; *Buckingham et al., 2011*; *Ishimoto et al., 2011*; *Yae et al., 2012*; *Timmerman et al., 2013*); inhibition of system $x_c^-$ may prove useful in a number of therapeutic contexts.

Efforts to treat gliomas and lymphomas in human patients by modulating system $x_c^-$ activity with the low potency, metabolically unstable small molecule, sulfasalazine (SAS, *Gout et al., 2001*), were

**eLife digest** Sugars, fats, amino acids, and other nutrients cannot simply diffuse into the cell. Rather, they must be transported across the cell membrane by specific proteins that stretch from one side of the cell membrane to the other. One such 'transporter'—system $x_c^-$—is of special interest. This transporter imports one molecule of cystine from outside the cell in exchange for one molecule of glutamate from inside the cell. Cystine, a variant of the amino acid cysteine, is essential for synthesizing new proteins and for preventing the accumulation of toxic species inside the cell. Not surprisingly, many cancer cells are dependent upon the transport activity of system $x_c^-$ for growth and survival. Drugs that can inhibit system $x_c^-$ could therefore be part of potential treatments for cancer and other diseases.

Dixon, Patel, et al. have found that the compound erastin is a very effective inhibitor of system $x_c^-$ function. Certain versions of erastin are over 1000 times more potent than the previously known best inhibitor of system $x_c^-$, sulfasalazine. Dixon, Patel et al. found that using erastin and sulfasalazine to inhibit system $x_c^-$ in cancer cells grown in a petri dish results in an unusual type of iron-dependent cell death called ferroptosis. By inhibiting the uptake of cystine, erastin and other system $x_c^-$ inhibitors interfere with the cellular machinery that folds proteins into their final, three-dimensional shape. The accumulation of these partially-folded proteins in the cell causes a specific kind of cellular stress that can be used as a readout, or biomarker, for the inhibition of system $x_c^-$. Such a biomarker will be essential for identifying cells in the body that have been exposed to agents that inhibit system $x_c^-$ and that are undergoing ferroptosis.

Unexpectedly, Dixon, Patel et al. also found that the FDA-approved anti-cancer drug sorafenib inhibits system $x_c^-$. Other drugs in the same class as sorafenib do not share this unusual property. Dixon, Patel et al. synthesized variants of sorafenib and identified sites on the drug that are necessary for it to be able to interfere with system $x_c^-$. Alongside the erastin derivatives, these new molecules may help to develop new drugs that can inhibit this important transporter in a clinical setting.

unsuccessful (*Robe et al., 2009*). While some progress has been made toward developing more potent compounds based on the SAS scaffold (*Shukla et al., 2011*), the identification of system $x_c^-$ inhibitors based on alternative scaffolds remains a pressing need and would be useful to test the hypothesis that system $x_c^-$ inhibition is therapeutically beneficial in glioma and other contexts. We previously demonstrated that the small molecule erastin prevents $Na^+$-independent cystine uptake (*Dixon et al., 2012*), suggesting that erastin may inhibit system $x_c^-$ function and represent a novel scaffold targeting this transport system. Intriguingly, treatment of some cell lines with erastin or SAS triggers an iron-dependent, non-apoptotic form of cell death, termed ferroptosis (*Dixon et al., 2012*). Ferroptosis is characterized by the accumulation of intracellular soluble and lipid reactive oxygen species (ROS), a process that is counteracted by the glutathione-dependent activity of the enzyme glutathione peroxidase 4 (GPX4) (*Dixon and Stockwell, 2013*; *Yang et al., 2014*). Erastin, and other ferroptosis-inducing compounds of this class, are therefore of interest both for their effects on amino acid transport and their ability to induce a novel cell death pathway.

In this study, we show that erastin and its analogs specifically inhibit cystine uptake via system $x_c^-$, trigger ferroptosis in a variety of cellular contexts and act much more potently than SAS. Surprisingly, we found that the clinically approved multi-kinase inhibitor sorafenib can also inhibit system $x_c^-$ and trigger ferroptosis under some conditions, an observation that may be relevant to both the anti-cancer properties and the profile of adverse events associated with this drug. We further show that small molecule inhibition of system $x_c^-$ function leads to endoplasmic reticulum (ER) stress, as indicated by the transcriptional upregulation of genes linked to the ER stress response. The upregulation of the ER stress response gene *CHAC1* (ChaC, cation transport regulator homolog 1) serves as a useful pharmacodynamic marker of system $x_c^-$ inhibition. Finally, we found that resistance to system $x_c^-$ inhibition is correlated with dramatically increased expression of *AKR1C* family members that regulate the detoxification of oxidative lipid breakdown products, providing potential insight into the downstream consequences of system $x_c^-$ inhibition, and the execution mechanism of ferroptosis.

## Results

### Consistent induction of ferroptosis in various cells under a variety of growth conditions

Erastin and SAS were previously shown to trigger ferroptosis in human HT-1080 fibrosarcoma cells grown on two-dimensional substrates with atmospheric levels of oxygen (i.e., 21% oxygen) (*Dixon et al., 2012*). We endeavored to generalize and validate the lethality of erastin towards cancer cells in several ways. First, we tested whether the same effects were observed in other cell types using a 'modulatory profiling' strategy (*Wolpaw et al., 2011*; *Dixon et al., 2012*). This method allows for the simplified detection and presentation of small molecule combination effects on cell viability (modulatory effect, $M_e < 0$, sensitization; $M_e = 0$, no effect; $M_e > 0$, rescue). We observed that in five different human cancer cell lines, cell death induced by either erastin or SAS was rescued by the same canonical ferroptosis inhibitors: the iron chelator ciclopirox olamine (CPX), the lipophilic antioxidants trolox and ferrostatin-1 (Fer-1), the MEK inhibitor U0126, the protein synthesis inhibitor cycloheximide (CHX) and the reducing agent beta-mercaptoethanol (β-ME) (*Dixon et al., 2012*; *Figure 1A,B*). Thus, the ferroptotic death phenotype, whether induced by erastin or SAS, was similar in all cell lines tested. The inhibition of cell death by β-ME indicates that cell death most likely involves inhibition of system $x_c^-$ function, as β-ME treatment can generate mixed disulfides taken up by other transporters, thereby circumventing the need for system $x_c^-$ function (*Ishii et al., 1981*).

Next, we sought to test whether the lethal mechanisms of action of erastin and SAS were influenced by cell growth architecture. Specifically, we tested whether the ferroptotic lethal mechanism could be activated in multicellular tumor spheroids (MCTSs), three-dimensional cellular aggregates proposed to recapitulate key aspects of the structural and metabolic heterogeneity observed in tumor fragments and micrometastases (*Friedrich et al., 2009*). We grew MCTSs from HT-1080 and Calu-1 cells for 72 hr and then investigated the effects of erastin ±β-ME or ±Fer-1 on MCTS growth and viability. For comparison, we also tested the growth inhibitory effects of (1*S*, 3*R*)-RSL3 (hereafter RSL3), a small molecule that triggers ferroptosis by inhibiting GPX4, which is downstream of system $x_c^-$ in the ferroptotic cascade (*Yang et al., 2014*), as well as staurosporine (STS), which triggers apoptosis. We observed that HT-1080 and Calu-1 MCTSs were killed by erastin and RSL3 (*Figure 1C,D*). The effects of both erastin and RSL3 were rescued by Fer-1, while β-ME suppressed the lethality of erastin, but not of RSL3, as expected (*Figure 1C,D*). Neither β-ME nor Fer-1 modulated the effects of STS on MCTS growth or viability (*Figure 1C,D*). These observations indicate that erastin, as well as RSL3, are able to trigger ferroptosis in a similar manner in both two- and three-dimensional culture conditions.

Finally, given that erastin triggers an oxidative form of cell death, we tested whether the lethality of erastin was affected by growth in low (1%) vs high (21%) levels of $O_2$. Cells from two different erastin-sensitive cancer cell lines (HT-1080 and DU-145) were grown for 24 hr under low or high $O_2$ levels and then treated for a further 24 hr with various agents, prior to the analysis of cell death. We observed that compared to DMSO-treated cells, erastin (5 μM)-treated cells were killed under both high and low $O_2$ conditions with little (DU-145) or no (HT-1080) difference in lethality (*Figure 1E,F*). In both cell lines, erastin-induced death was suppressed by both Fer-1 (1 μM) and CPX (5 μM) (*Figure 1E,F*), indicating that the same lethal mechanism (i.e., ferroptosis) was responsible for cell death under both high and low $O_2$ conditions. Thus, even under relatively low $O_2$ conditions, it is still possible for erastin to activate the ferroptotic mechanism.

### Erastin inhibits system $x_c^-$ function potently and specifically

The ability to modulate system $x_c^-$ activity may be clinically useful, but requires small molecule inhibitors with suitable pharmacological properties that are also specific for this antiporter (*Gorrini et al., 2013*). Erastin treatment (5 μM) completely abolished the $Na^+$-independent uptake of radiolabelled [$^{14}$C]-cystine in both HT-1080 fibrosarcoma and Calu-1 lung carcinoma cancer cells, as did sulfasalazine (SAS) at 100-fold higher concentrations (500 μM) (*Figure 2A*). Conversely, erastin and SAS had no effect on $Na^+$-independent [$^{14}$C]-phenylalanine uptake (*Figure 2B*). An excess of cold D-phenylalanine did suppress [$^{14}$C]-phenylalanine uptake, confirming that Phe transport was inhibitable under these assay conditions (*Figure 2B*). Thus, in HT-1080 and Calu-1 cells, erastin and SAS block system $x_c^-$ (SLC7A11 + SLC3A2)-mediated cystine uptake selectivity over other transport systems and amino acids, such as system-L-(SLC7A5 + SLC3A2)-mediated Phe uptake.

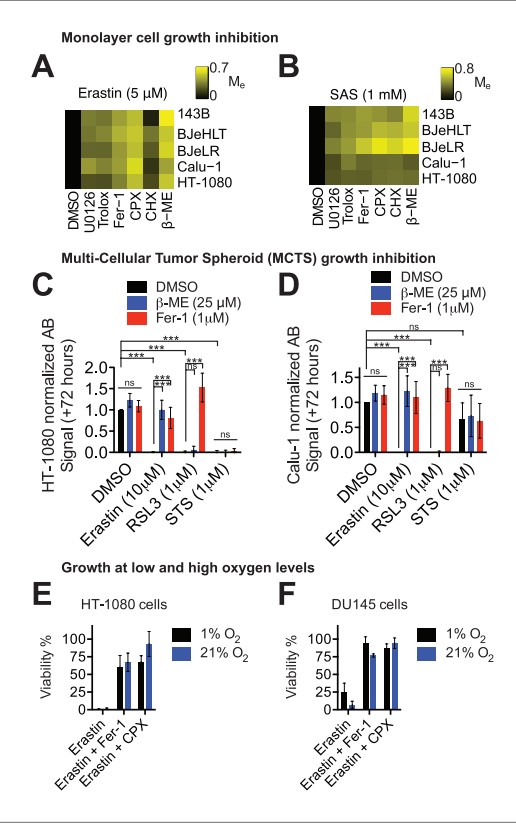

**Figure 1**. Cell death is triggered by erastin and related compounds in different cell lines under a variety of physiological conditions. (**A** and **B**) Modulatory effect ($M_e$) profiles of erastin- and SAS-induced death in five different cell lines (143B, BJeHLT, BJeLR, Calu-1, and HT-1080) in response to six different cell death inhibitors (U0126, Trolox, Fer-1, CPX, CHX, β–ME) or the vehicle DMSO. $M_e$ >0 indicates rescue from cell death. (**C** and **D**) Relative viability of MCTSs formed over 72 hr from HT-1080 (**C**) or Calu-1 (**D**) cells in response to erastin, RSL3 or staurosporine (STS) ±β–ME or ferrostatin-1 (Fer-1). Viability was assessed by Alamar blue and represents mean ± SD from three independent biological replicate experiments. Data were analyzed by two-way ANOVA with Bonferroni post-tests, *p<0.05, **p<0.05, ***p<0.001, ns = not significant. (**E** and **F**) Viability of HT-1080 (**E**) and DU145 (**F**) cells cultured under 1% or 21% $O_2$ levels in response to erastin (5 µM) ±Fer-1 (1 µM) or CPX (5 µM). Viability was assessed by Alamar blue and represents mean ± SD from three independent biological replicate experiments.

We confirmed the ability of erastin and SAS to inhibit system $x_c^-$ using an enzyme-coupled fluorescent assay that detects glutamate release into Na$^+$-containing culture medium (***Figure 2—figure supplement 1A***). We validated this assay in three ways. First, we showed that erastin (**1**) inhibited glutamate release, while a non-lethal (***Yagoda et al., 2007***) erastin analog lacking the p-chlorophenoxy moiety (erastin-A8, **2**) did not (***Figure 2C,D***). Second, we showed that both erastin treatment and silencing of *SLC7A11* with either of two independent siRNAs resulted in a significant, quantitatively similar inhibition of glutamate release (***Figure 2E,F***); silencing of the system L transporter subunit *SLC7A5* using two independent siRNAs had no effect on basal or erastin-mediated inhibition of glutamate release (***Figure 2—figure supplement 1B,C***). Third, we found that only erastin and SAS inhibited glutamate release, while, as expected, RSL3, artesunate and PEITC did not; while artesunate and PEITC induce iron-dependent cell death, neither are known to inhibit system $x_c^-$ or induce ferroptosis (***Trachootham et al., 2006***; ***Hamacher-Brady et al., 2011***; ***Dixon et al., 2012***; ***Figure 2G***). Thus, the above results suggest that both erastin and SAS specifically inhibit SLC7A11-dependent system $x_c^-$ function. The ability of erastin to specifically inhibit cystine uptake via system $x_c^-$ is further supported by recent metabolomic profiling data (***Skouta et al., 2014***; ***Yang et al., 2014***) and gene expression experiments described below.

In light of disappointing clinical results using SAS (***Robe et al., 2009***), it is desirable to identify potent inhibitors of system $x_c^-$ with favorable pharmacological properties. Using the glutamate release assay to quantify inhibition of system $x_c^-$ activity, we found that erastin was ~2500 times more potent than SAS as an inhibitor of system $x_c^-$ function in both HT-1080 and Calu-1 cells (HT-1080: erastin IC$_{50}$ = 0.20 µM, 95% C.I. 0.11–0.34 µM, SAS IC$_{50}$ = 450 µM, 95% C.I. 280–710 µM; Calu-1: erastin IC$_{50}$ = 0.14 µM, 95% C.I. 0.081–0.21 µM, SAS IC$_{50}$ = 460 µM, 95% C.I. 350–590 µM) (***Figure 2H***). Thus, the erastin scaffold may afford a more suitable starting point than SAS for the development of potent and selective inhibitors of system $x_c^-$ function.

## Erastin structure-activity relationship (SAR) analysis and isolation of analogs with improved potency

We hypothesized that it would be possible to improve further the potency of the erastin scaffold through targeted synthesis. Towards this end, we undertook a search for more potent analogs, beginning with an achiral analog (**3**, ***Yang et al., 2014***) that lacked the methyl group at the chiral center, and that had an isoproproxy substituent in place of the ethoxy group on erastin (**1**) (***Figure 3A***). This compound (**3**) was more synthetically accessible, but otherwise exhibited a similar lethal potency as erastin in

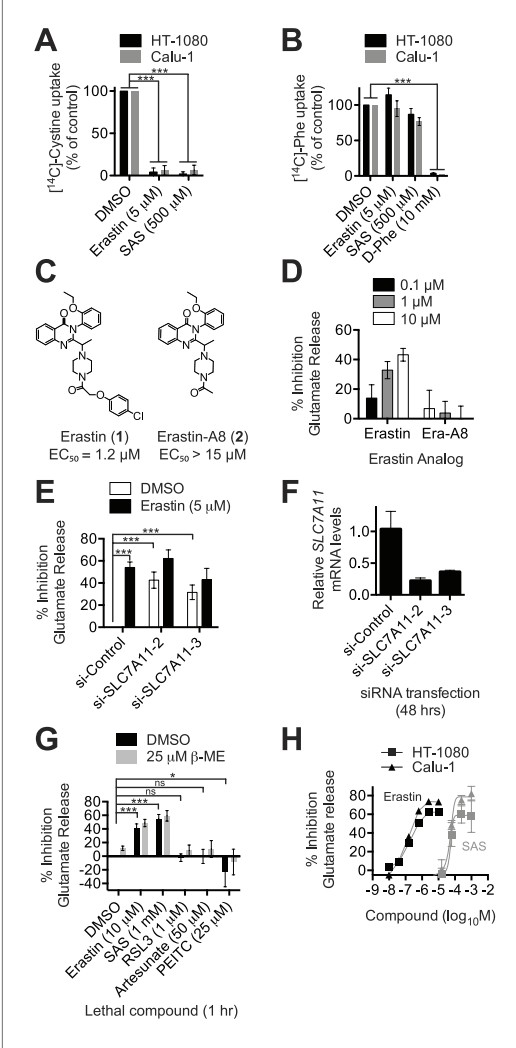

**Figure 2**. Erastin inhibits system $x_c^-$ function potently and specifically. (**A** and **B**) Na$^+$-independent uptake of $^{14}$C-cystine (**A**) and $^{14}$C-L-phenylalanine (Phe) (**B**) over 5 min in HT-1080 and Calu-1 cells treated with erastin or SAS. D-Phe was included as a positive control in **B**. (**C**) Structure and lethal potency (EC$_{50}$ in HT-1080 cells) of erastin and the inactive erastin analog erastin-A8. (**D**) Dose-dependent inhibition of glutamate release by erastin and erastin-A8 (Era-A8). (**E**) Glutamate release ±erastin in HT-1080 cells in which *SLC7A11* was silenced for 48 hr using two independent siRNAs. (**F**) *SLC7A11* mRNA levels assayed using RT-qPCR in si-*SLC7A11*-transfected cells. (**G**) Glutamate release in response to erastin, SAS, RSL3 artesunate and PEITC, ±beta-mercaptoethanol (β-ME). (**H**) Dose-response analysis of glutamate release from HT-1080 and Calu-1 cells in response to erastin and SAS. All data are from three independent biological replicates. Data are presented as mean ± SD. Data in **A** and **B** are normalized to DMSO controls (set to 100%). Data in **A**, **B**, **E** and **G** were analyzed by ANOVA with Bonferroni post-tests, *p<0.05, ***p<0.001, ns = not significant.

*Figure 2. Continued on next page*

HT-1080 cells. We synthesized 19 analogs of **3** (*Supplementary file 1*, *Figure 3A*. Data also available as 'Extended Materials and Methods' from Dryad data Repository [*Dixon et al., 2014*]), and tested each in HT-1080 cells in a 10-point, twofold dose-response assay for lethal potency and efficacy (*Figure 3B*). To assess in each case whether lethality involved inhibition of system $x_c^-$ and induction of ferroptosis, as opposed to the induction of another form of death, experiments were performed ±β-ME. To assess in a parallel assay the correlation between lethal potency and inhibition of system $x_c^-$ activity, we examined glutamate release using a high throughput, 96-well Amplex Red assay system in human CCF-STTG1 astrocytoma cells (*Figure 3B*). Overall, the lethal potency in HT-1080 cells was found to correlate significantly with the degree of system $x_c^-$ inhibition observed in CCF-STTG1 cells (Pearson $R^2 = 0.86$, p<0.0001). Of note, we observed that glutamate release in CCF-STTG1 cells was in general more sensitive to erastin and analogs than HT-1080 cells (compare *Figure 3B* to *Figure 2H*). These results support the hypothesis that the ability of erastin analogs to trigger ferroptosis is quantitatively linked to their ability to inhibit system $x_c^-$ function.

We investigated the above data set in more detail for insights into the erastin structure activity relationship. Erastin's quinazolinone core (Region A) is found in a number of biologically active compounds and is considered to be a 'privileged' scaffold (*Welsch et al., 2010*). Modifications to this region (**4**–**10**), including substitution of the quinazolinone for quinolone (**4**) or indole (**5**), obtained using a Meth-Cohn quinoline synthesis (*Supplementary file 1*), resulted in moderate to severe losses of lethal potency compared to **3**, suggesting that the quinazolinone core scaffold is essential for the lethality of erastin. Modifications to the piperazine linker (Region B, **11**–**12**) were not tolerated, with **12** being completely inactive in both the HT-1080 lethality and CCF-STTG1 glutamate release assays. We speculate that rigidification of this portion of the scaffold is essential for activity and that an increase in the number of rotatable bonds in this region results in a higher entropic cost of binding, decreasing lethal potency. Single atom changes to the acetoxy spacer (Region C, **13**–**15**) were likewise poorly tolerated, resulting in significant losses in potency that correlated with reduced inhibition of system $x_c^-$ activity. Strikingly, subtle modifications to Region D, including replacement of the chlorine with a fluorine (**16**), replacement of the *para*-chloro substituent with a *meta*-chloro

*Figure 2. Continued*

The following figure supplements are available for figure 2:

**Figure supplement 1**. Monitoring system $x_c^-$ activity by following glutamate release.

(**17**) or elimination of it altogether (**18**) reduced or abrogated both lethality and system $x_c^-$ inhibitory activity. As suggested by the weakened potency of **16** and **17**, and the inactivity of **18**, the chlorine atom may make a key halogen bonding interaction with the surrounding environment (*Wilcken et al., 2012*) that is essential for binding.

Finally, modifications to Region E, including addition of a bromo group (**20**), a phenyl (**21**) or a furanyl substituent (**22**) resulted in fivefold or greater improvements in lethal potency that were mirrored by fivefold to 10-fold improvements in the inhibition of system $x_c^-$ activity compared to **3**; the most potent compound, **21**, inhibited glutamate release with below 5 nM potency in the CCF-STTG1 assay. Crucially, these more potent compounds potently triggered lethality in HT-1080 cells via ferroptosis, as death was fully suppressed by β-ME. Previously, we have shown that erastin and lethal analogs thereof demonstrate selective lethality towards human BJ fibroblasts engineered to express human telomerase, SV40 large and small T antigen, and oncogenic $HRAS^{V12}$ (BJeLR) compared to isogenic cells expressing only telomerase (BJeH) (*Dolma et al., 2003*; *Yang et al., 2014*). We tested the most potent lethal analog (**21**), along with the parent compound (**3**) and a representative non-lethal analog (**14**), in these cell lines. While **14** was inactive, we found that both lethal analogs (**21** and **3**) retained selectivity towards BJeLR vs BJeH cells (*Figure 3C*). Consistent with the pattern of lethality observed in HT-1080 cells, **21** was a more than 20-fold more potent lethal molecule compared to **3** (BJeLR $EC_{50}$ of 22 nM [95% C.I. 20-25 nM] vs 490 nM [95% C.I. 350-690 nM], respectively). Together, these results demonstrate that it is possible to improve the potency of the erastin scaffold substantially while retaining oncogenic RAS-selective lethality, especially via modifications to Region E. The combination of these new structural insights, together with complementary results concerning modifications that enhance the metabolic stability of erastin (*Yang et al., 2014*), may result in suitable compounds for clinical studies.

## The effects of erastin on the transcriptome are due to depletion of cystine

Given the above results, we hypothesized that the effects of erastin were due entirely to inhibition of system $x_c^-$ function and the consequent depletion of cystine (and ultimately cysteine) from the intracellular milieu. If so, co-treatment with β-ME should reverse all effects resulting from erastin treatment. To test this hypothesis in a global manner, we examined patterns of changes in the transcriptome using RNA sequencing (RNA-Seq) of mRNA harvested from HT-1080 cells treated for 5 hr with DMSO, erastin (10 μM), β-ME (18 μM) or erastin + β-ME. From two independent biological replicates, we obtained an average of ~30.5 million unique mapped reads and 11,867 unique transcripts (with Fragments Per Kilobase of exon per Million reads [FPKM] ≥1 in both replicates) per condition. After data processing and averaging of replicates, we identified 33 mRNAs with two-fold more counts ('up-regulated') and four mRNAs with twofold fewer counts ('down-regulated') in erastin-treated samples vs DMSO-treated controls (*Figure 4A,B*; Data available as 'Data Package 1' from Dryad data Repository [*Dixon et al., 2014*]). In support of the hypothesis, erastin-induced changes in mRNA expression were reversed by co-treatment with β-ME for all 33 up-regulated genes (Mann–Whitney test, $p<0.0001$) and for each of the four down-regulated genes. These results suggest that the effects of erastin on cellular physiology detectable at the level of mRNA expression are due to depletion of intracellular cystine, arising as a consequence of inhibition of system $x_c^-$ function.

## Erastin triggers an endoplasmic reticulum (ER) stress

We noted that several of the genes upregulated by erastin were associated with activation of the eIF2alpha-ATF4 branch of the ER stress response pathway (e.g., *ATF3*, *DDIT3*, *DDIT4* [*Jiang et al., 2004*; *Whitney et al., 2009*]). Consistent with this, we observed that the 33 up-regulated genes were significantly enriched for GO Biological Process terms related directly to the ER stress and unfolded protein responses (GO:0034976, response to endoplasmic reticulum stress, $p=8.0\ e^{-11}$; GO:0006987, activation of signaling protein activity involved in unfolded protein response, $p=1.3\ e^{-9}$; GO:0032075, positive regulation of nuclease activity, $p=1.5\ e^{-9}$). The eIF2alpha-ATF4 branch of the ER stress/unfolded protein response can be upregulated by amino acid depletion (*Harding et al., 2003*), which we hypothesize is linked to intracellular cysteine depletion downstream of system $x_c^-$ inhibition by

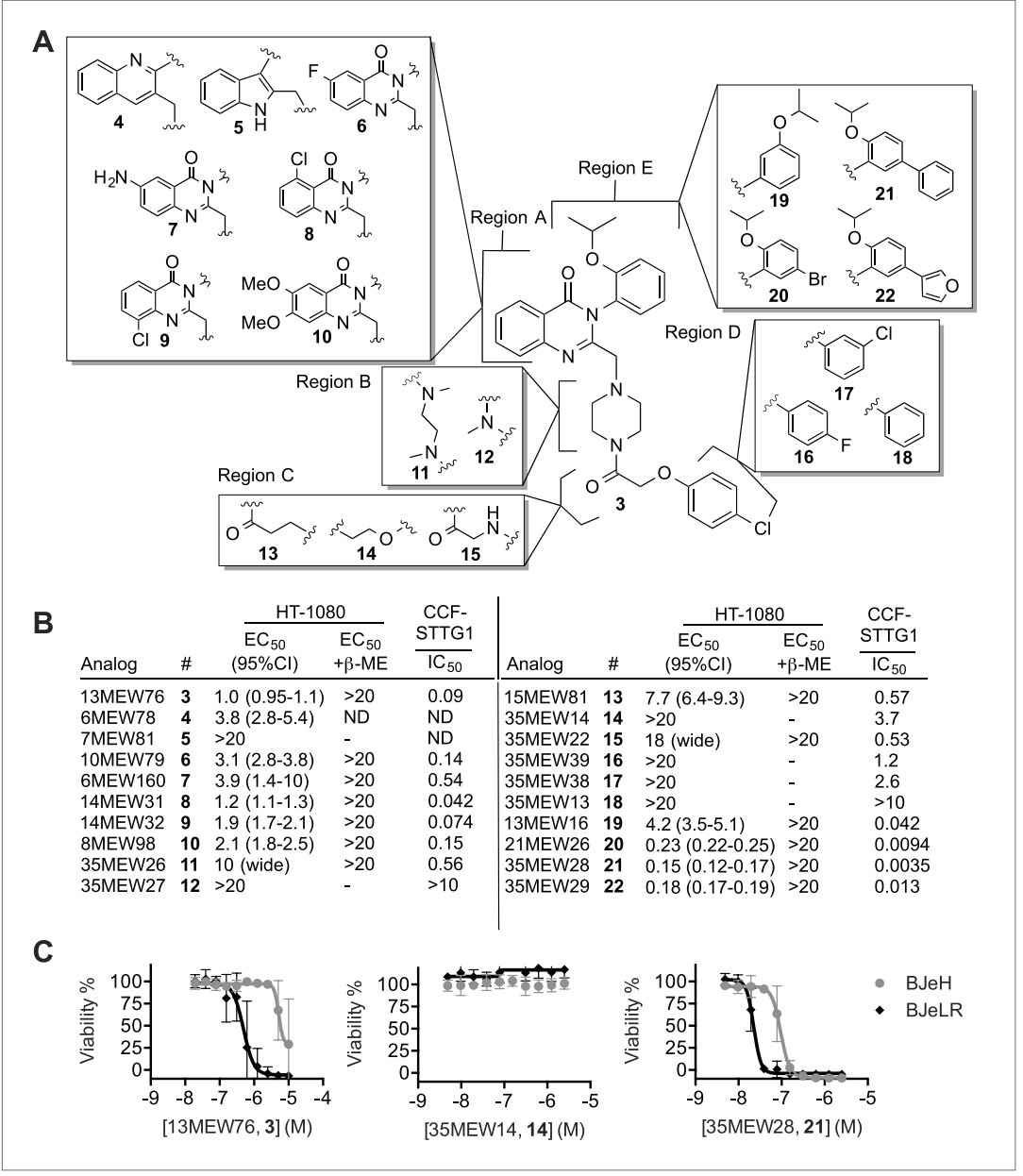

**Figure 3**. Structure activity relationship (SAR) analysis of erastin. (**A**) Structures of 20 erastin analogs. (**B**) Lethal $EC_{50}$ for each analog determined in HT-1080 cells in a 10-point, twofold dilution assay, starting at a high concentration of 20 μM, ±β-ME (18 μM). Data represent mean and 95% confidence interval (95% C.I.) from three independent biological replicate experiments. Also reported are $IC_{50}$ values for inhibition of glutamate release as determined in CCF-STTG1 cells. These data represent the average of two experiments. All values are in μM. ND: not determined. (**C**) Dose-response curves for selected erastin analogs (**3**, **14** and **21**) in BJeLR and BJeH cells. Data represent mean ± SD from three independent biological replicates.

erastin. We investigated further the connection between erastin treatment and activation of the eIF2alpha-ATF4 pathway and observed that, relative to DMSO-treated controls, erastin treatment (5 μM, 7 hr) resulted in phosphorylation of eIF2alpha and up-regulation of ATF4 at the protein level (**Figure 4—figure supplement 1A**). We saw no evidence for enhanced splicing of the XBP1 mRNA, which provides a readout for activation of a parallel ER stress response pathway (**Figure 4—figure supplement 1B**). In HT-1080 cells, co-treatment with the transcriptional inhibitor actinomycin D (1 μg/ml) inhibited erastin-induced changes in gene expression (see below) and delayed but did not prevent

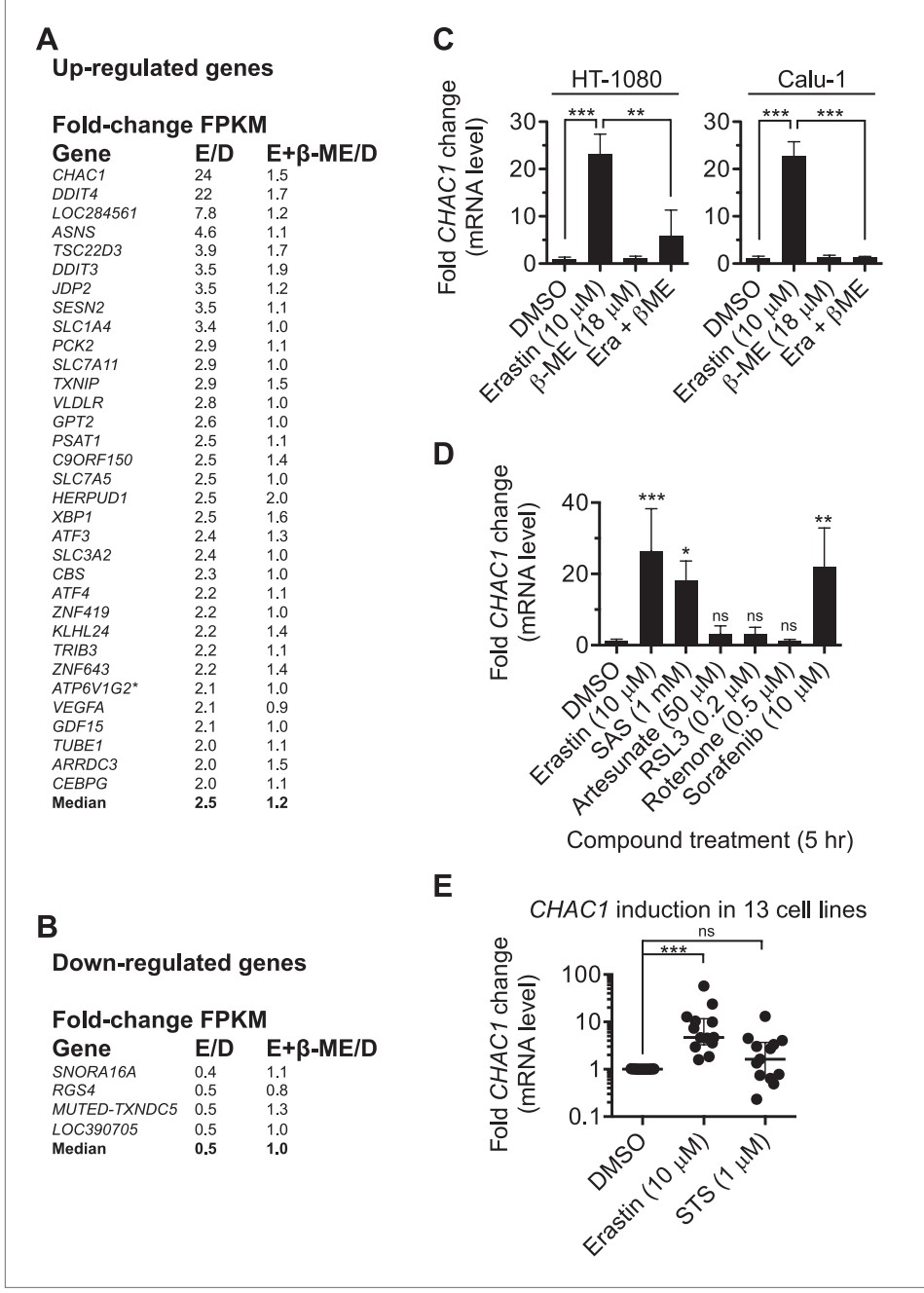

**Figure 4**. Analysis of erastin effects using RNA-Seq. (**A** and **B**) List of genes upregulated (**A**) and downregulated (**B**) by erastin treatment, as detected in HT-1080 cells using RNA Seq. The number of fragments per kilobase of exon per megabase of sequence (FPKM) was counted and is expressed as a fold-change ratio between the different conditions. E/D: Erastin/DMSO expression ratio. E+β-ME/D: Erastin+β-ME/DMSO ratio. *ATP6V1G2*\*: *ATP6V1G2-DDX39B* read-through transcript. Data represent the average of two independent biological replicates for each condition. (**C** and **D**) mRNA expression level of *CHAC1* determined by RT-qPCR in HT-1080 and Calu-1 cells in response to erastin ±β-ME treatment for 5 hr. Data are from three independent biological replicates and represented as mean ± SD and were analyzed by one-way ANOVA with Bonferroni post-tests, \*\*p<0.01, \*\*\*p<0.001, ns = not significant. In **D** significance is indicated relative to the DMSO control. (**E**) *CHAC1* mRNA levels in 13 different erastin-sensitive cell lines treated with erastin or STS (6 hr). Results in **E** were analyzed using the Kruskal–Wallis test, \*\*\*p<0.001, ns = not significant.

The following figure supplements are available for figure 4:

**Figure supplement 1**. Analysis of ER stress in response to erastin treatment.

erastin-induced cell death (*Figure 4—figure supplement 1C*). Thus, while blockade of system $x_c^-$ by erastin (and other agents, see below) can trigger a robust transcriptional signature indicative of ER stress, it is doubtful that this transcriptional response is essential for the lethality observed following erastin treatment.

## Transcriptional changes reveal a pharmacodynamic marker for erastin exposure

Pharmacodynamic (PD) markers would be useful to determine when cells are responding to system $x_c^-$ inhibition, such as in response to erastin treatment. We therefore explored the RNA-Seq profiles for suitable candidate PD makers; the most highly up-regulated gene observed in erastin-treated HT-1080 cells by RNA-Seq was *CHAC1* (~24-fold, *Figure 4A*), an ER stress-responsive gene known to be upregulated downstream of ATF4 (*Gargalovic et al., 2006*; *Mungrue et al., 2009*). We validated these results by RT-qPCR using fresh samples prepared from HT-1080 and Calu-1 cells, and confirmed that *CHAC1* up-regulation was fully reversible by co-treatment with β-ME (*Figure 4C*). *CHAC1* mRNA upregulation was observed in response to seven different active erastin analogs described above (**3**, **20**–**22**) and in a recent publication (AE, PE and MEII, [*Yang et al., 2014*]); low levels of *CHAC1* upregulation were also observed in response to two less potent analogs (**14**, **16**, both of which nonetheless retain some ability to inhibit system $x_c^-$ function, see *Figure 3B*), suggesting that the induction of ER stress and *CHAC1* upregulation may be more sensitive to the inhibition of system $x_c^-$ than cell viability (*Figure 4—figure supplement 1E,F*). We examined the specificity of the above response—we observed transcriptional upregulation of *CHAC1* following treatment with system $x_c^-$ inhibitors (erastin and SAS), but not in response to RSL3, artesunate, rotenone or buthionine sulfoximine (BSO), agents that induce oxidative stress, but that do not inhibit system $x_c^-$ (*Figure 4D*, *Figure 4—figure supplement 1D*), suggesting that *CHAC1* upregulation can specifically indicate agents that inhibit system $x_c^-$ function vs those that trigger redox stress by other means. Upregulation of *CHAC1* in erastin-treated HT-1080 cells was prevented by co-treatment with the transcriptional inhibitor actinomycin D as well as the protein synthesis inhibitor CHX, suggesting that *CHAC1* mRNA upregulation downstream of erastin treatment requires new transcription and translation (*Figure 4—figure supplement 1D*).

We next tested the generality of *CHAC1* upregulation in response to erastin, and observed that across a panel of 13 cancer cell lines, treatment with erastin, but not the apoptosis-inducer STS, resulted in a significant increase in *CHAC1* expression (*Figure 4E*). Thus, erastin can trigger a number of changes in cell physiology specifically linked to cystine depletion, and *CHAC1* up-regulation could be a useful transcriptional PD marker for exposure to erastin and other agents that deplete cells of cystine or cysteine. This may be useful in testing other potential inhibitors of system $x_c^-$ function.

## Modulatory profiling identifies sorafenib as an inhibitor of system $x_c^-$

Inhibition of system $x_c^-$ activity and/or glutathione depletion may be useful in combination with other therapies to selectively target specific tumor types or sensitize them to other agents (*Dai et al., 2007*; *Ishimoto et al., 2011*; *Timmerman et al., 2013*). We therefore used a modulatory profiling strategy to test whether the lethality of 20 mechanistically diverse compounds could be enhanced in both A549 and HCT-116 cells by system $x_c^-$ inhibition using erastin (10 μM), or by glutathione depletion using BSO (2.5 mM). Overall, we observed that the modulatory effect ($M_e$) values for the test compounds clustered around zero (i.e., additive enhancement of death), with the exception of RSL3, phenylarsine oxide (PAO) and sorafenib (*Figure 5A,B*). RSL3 induces ferroptosis through a mechanism independent of system $x_c^-$ (see above), while PAO binds vicinal thiols and has lethal activity that is opposed by glutathione-dependent enzymes (*Lillig et al., 2004*), rationalizing the synergistic effects observed with these two compounds. The observation that sorafenib (BAY 43-9006, Nexavar), a multi-kinase inhibitor clinically approved for the treatment of renal cell carcinoma (*Wilhelm et al., 2006*), could synergize with erastin or BSO treatment was unexpected, and suggested that sorafenib could affect the ferroptosis pathway.

We investigated this hypothesis as follows. First, we found that in HT-1080 cells sorafenib (10 μM, 24 hr) treatment-induced cell death was significantly inhibited by the known small molecule ferroptosis suppressors β-ME, Fer-1 and DFO (*Dixon et al., 2012*); these same inhibitors suppress erastin (10 μM, 24 hr)-induced ferroptotic death, but not STS-induced apoptotic death (*Figure 5C*). Thus, sorafenib alone can trigger ferroptosis. The observed suppression of sorafenib-induced death by β-ME

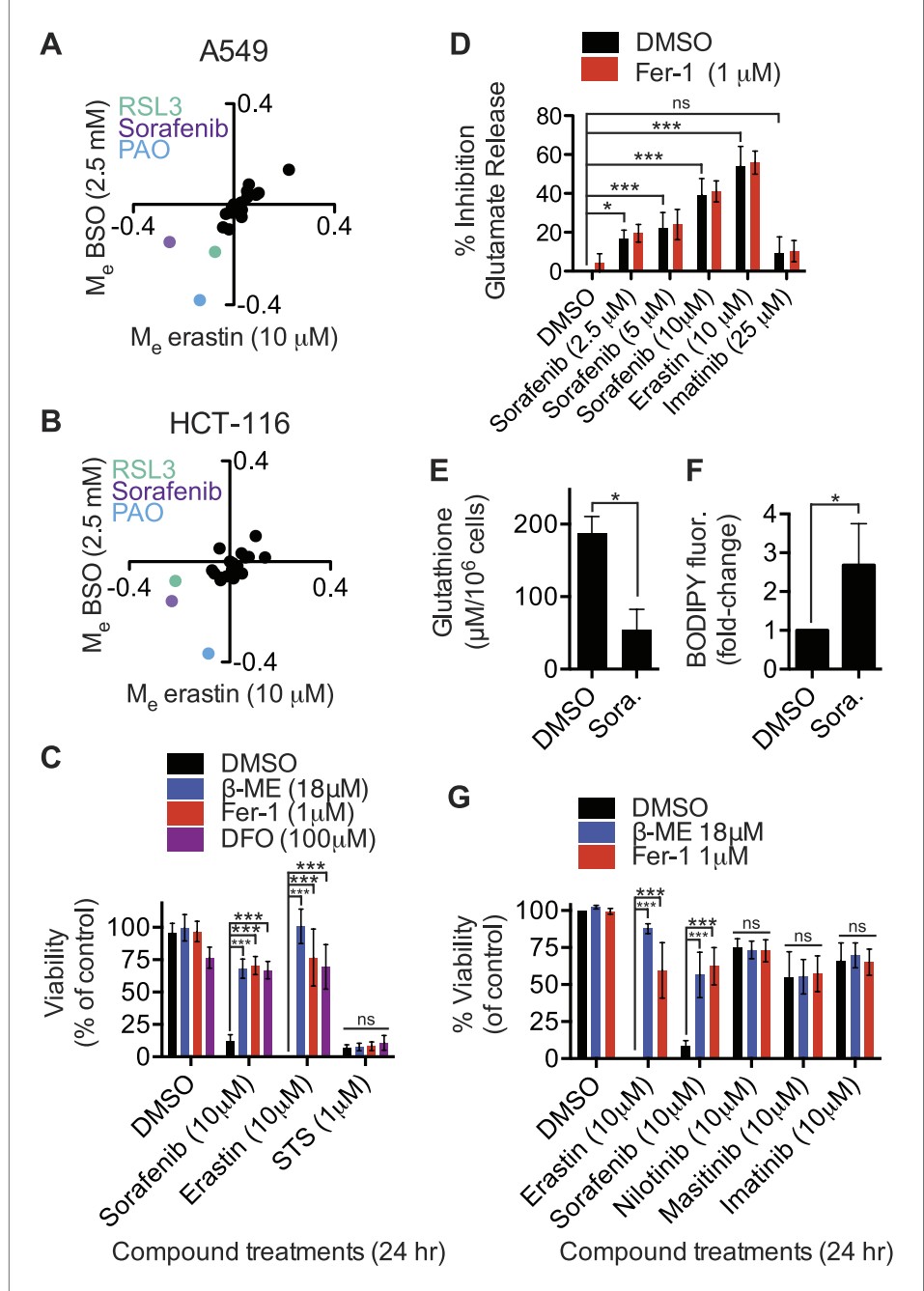

**Figure 5**. Identification of sorafenib as an inhibitor of system x$_c^-$. (**A and B**) Modulatory profiling of (**A**) A549 and (**B**) HCT-116 cells in response to either buthionine sulfoximine (BSO) or erastin ±20 different lethal compounds (see 'Materials and methods' for the full list). (**C**) Viability of HT-1080 cells treated for 24 hr with ferroptosis inhibitors (β-ME, Fer-1, DFO) ±sorafenib, erastin or STS. (**D**) Quantification of the inhibition of glutamate release by sorafenib, erastin and imatinib ±Fer-1. (**E and F**) HT-1080 cells treated with control vehicle (DMSO) or sorafenib (10 μM) for 18 hr prior to the assay. (**E**) Total glutathione levels measured using Ellman's reagent. (**F**) Lipid ROS levels assayed using C11-BODIPY 581/591. (**G**) Viability of HT-1080 cells treated for 24 hr with erastin, sorafenib, nilotinib, masitinib or imatinib ±β-ME or Fer-1. Data in **C**–**G** represent mean ± SD from at least three independent biological replicates. Cell viability in **C** and **G** was quantified by Alamar blue. Data in **C**, **D** and **G** were analyzed by one- and two-way ANOVA with Bonferroni post-tests; data in **E** and **F** were analyzed using Student's *t*test, *p<0.05, **p<0.01, ***p<0.001, ns = not significant relative to the indicated treatments. In (**D**), none of the comparisons between DMSO and Fer-1 treated samples were significant (p>0.05).

*Figure 5. Continued on next page*

*Figure 5. Continued*
The following figure supplements are available for figure 5:
**Figure supplement 1**. Effect of sorafenib on cell viability.

was striking and immediately suggested that sorafenib, like erastin and SAS, could be acting to inhibit system $x_c^-$ function. Indeed, we observed that sorafenib, like erastin, but not the kinase inhibitor imatinib, resulted in a dose-dependent inhibition of system $x_c^-$ function, as assessed using the glutamate release assay in HT-1080 cells (*Figure 5D*). The ability of sorafenib and erastin to suppress system $x_c^-$ activity was not inhibited by co-treatment with Fer-1, demonstrating that this effect is upstream of Fer-1-sensitive ROS accumulation (*Figure 5D*), as expected. Likewise, similar to erastin and SAS, we observed a robust transcriptional upregulation of *CHAC1* in HT-1080 cells in response to sorafenib treatment (*Figure 4D*). We also observed upregulation of the same biochemical markers of ER stress observed previously with erastin, namely phosphorylation of eIF2alpha and increased levels of ATF4, without any change in XBP1 splicing (*Figure 4—figure supplement 1A,B*). Finally, as observed previously with erastin treatment (*Dixon et al., 2012*; *Yang et al., 2014*), we found that sorafenib treatment (10 µM, 18 hr) of HT-1080 cells significantly depleted total glutathione and resulted in the accumulation of lipid peroxides as detected by flow cytometry using C11-BODIPY 581/591 (*Figure 5E,F*). Together, these results suggest that, like erastin, sorafenib inhibits system $x_c^-$-mediated cystine import, leading to ER stress, glutathione depletion and the iron-dependent accumulation of lipid ROS.

To test the generality of these results, we examined the ability of sorafenib to inhibit system $x_c^-$ activity and trigger ferroptosis in additional cell lines. Consistent with the initial results, in all five cell lines examined, we observed that sorafenib and erastin treatments (20 µM) caused a comparable inhibition of system $x_c^-$ function, as assessed by glutamate release (*Figure 5—figure supplement 1A*). However, unlike erastin, we observed that sorafenib triggered Fer-1-suppressible ferroptosis in HT-1080 cells only within a narrow concentration window (sorafenib $EC_{50}$ = 18 µM, sorafenib+Fer-1 $EC_{50}$ = 43 µM), before causing Fer-1-insensitive death at higher concentrations (*Figure 5—figure supplement 1B*). In the four other cells lines, we observed only slight (143B) or non-existent (TC32, Calu-1, U2OS) differences in sorafenib $EC_{50}$ values either with or without Fer-1 (*Figure 5—figure supplement 1B*). Thus, while sorafenib can inhibit system $x_c^-$ activity robustly, this manifests as a ferroptotic cell death phenotype only over a narrow range of concentrations; sorafenib appears to trigger additional lethal mechanisms that act in parallel to the ferroptotic response at higher concentrations.

## Analyzing the concordance between inhibition of system $x_c^-$ and lethality of sorafenib using structure activity relationship analysis

Sorafenib could conceivably inhibit system $x_c^-$ activity by modulating the activity of a kinase that controls system $x_c^-$ function, or through 'off-target' modulation of a non-kinase target (e.g., SLC7A11 itself, a system $x_c^-$ regulatory protein, or a more indirectly related target). We took two approaches in an attempt to address this question. First, we examined the effects of functionally-related kinase inhibitors. The global pattern of kinase inhibition by sorafenib against 300 purified kinase domains is highly similar to that of nilotinib, masitinib, and imatinib (*Anastassiadis et al., 2011*), yet none of these agents appear to trigger ferroptosis, as defined by sensitivity to ferroptosis-specific cell death inhibitors β-ME and Fer-1 (*Figure 5G*). Thus, even kinase inhibitors with targets similar to those of sorafenib do not necessarily trigger ferroptosis.

Second, we attempted to dissociate the ability of sorafenib to trigger ferroptosis vs other lethal mechanisms. To do this we synthesized and tested a set of 87 sorafenib analogs for the ability to trigger β-ME- and Fer-1-suppressible death in HT-1080 cells in twofold, 10-point dilution series assays, starting at a highest concentration of 20 or 40 µM. In summary, while many analogs retained lethal activity (three examples of lethal compounds are shown in *Figure 6A*), none of the 87 analogs could trigger ferroptosis with enhanced selectivity for β-ME- and Fer-1-suppressible death over other lethal mechanisms compared to the parent compound itself. Several of the analogs that we synthesized were unable to trigger death (four example compounds are shown in *Figure 6B*). These analogs (SRS13-67, SRS14-98, SRS13-48, and SRS15-11) contain modifications predicted to disrupt atomic interactions essential for the binding of sorafenib to kinase targets such as BRAF, including burial of the -CF3 group

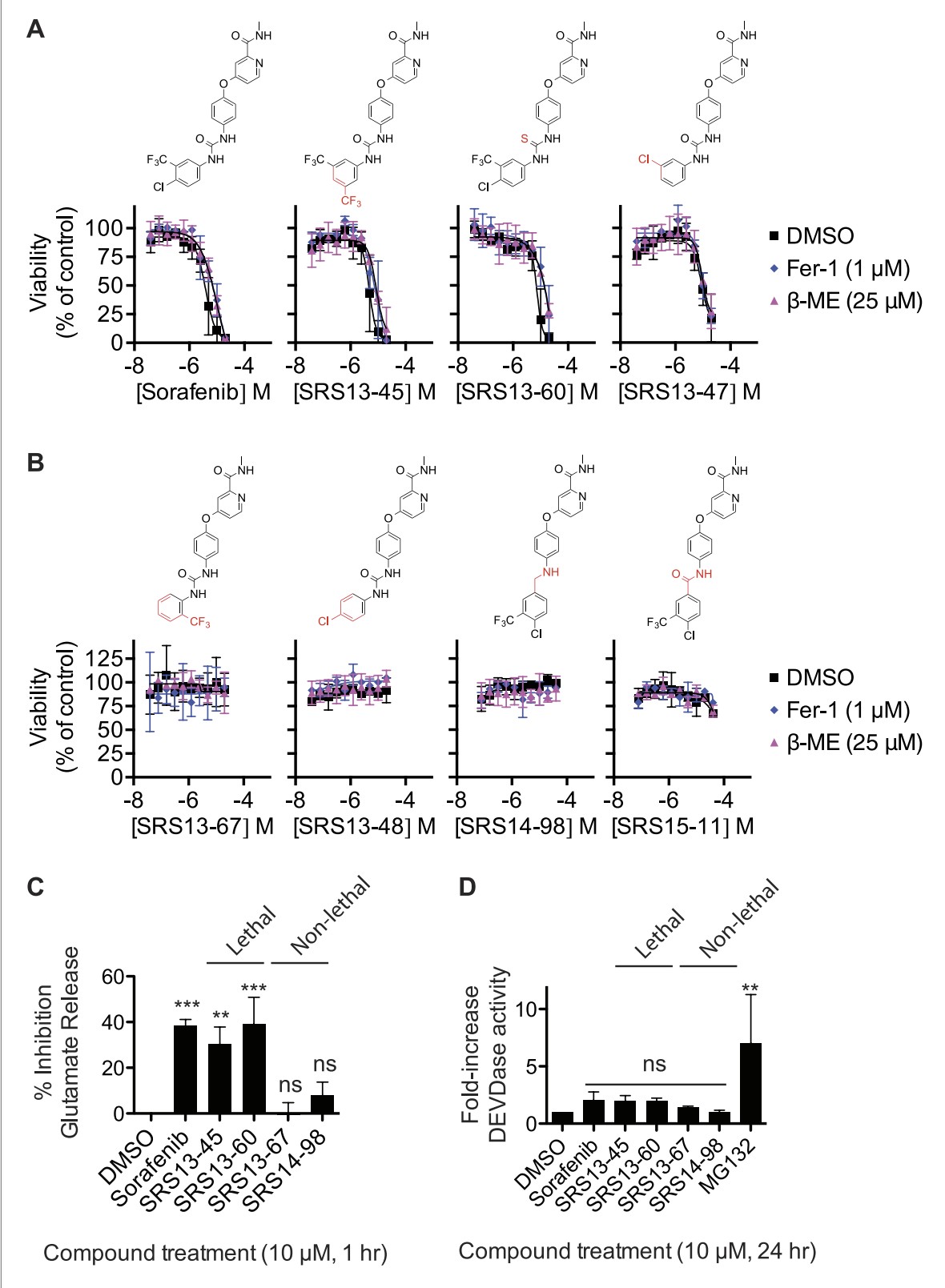

**Figure 6**. Analysis of sorafenib analog function. (**A** and **B**) Sorafenib and 87 sorafenib analogs were prepared and tested in HT-1080 cells for the induction of cell death (EC$_{50}$) and the suppression of this cell death by β-ME (18 μM) and Fer-1 (1 μM) over 48 hr. Representative data are shown for three analogs that retained lethal activity (SRS13-45, SRS13-60, SRS13-47) and four analogs that lost lethal activity (SRS13-67, SRS13-48, SRS14-98, SRS15-11) in *Figure 6. Continued on next page*

*Figure 6. Continued*

the cell-based assay. (**C**) Glutamate release in response to sorafenib and select analogs. (**D**) DEVDase (caspase-3/7) activity in response to sorafenib and select analogs as measured by the cleavage of a fluorescent rhodamine-linked substrate.

in a hydrophobic pocket and hydrogen bonding with the urea group (*Lowinger et al., 2002*; *Wan et al., 2004*). Conversely, the active analogs shown here (*Figure 6A*) mostly retain these features. Thus, one hypothesis is that sorafenib triggers ferroptosis by inhibiting an unknown kinase whose activity is necessary for constitutive system $x_c^-$ activity. Alternatively, sorafenib could modulate system $x_c^-$ activity by interacting with a non-kinase target that harbors a binding pocket resembling that found within the active site of sorafenib-sensitive kinases.

To further bolster the working model, we evaluated four of the above sorafenib analogs (two lethal, two non-lethal) for their effects on system $x_c^-$ function, using the glutamate release assay, and on the induction of apoptotic death, using a fluorogenic caspase-3/7 substrate cleavage assay. Consistent with the above data, two lethal sorafenib analogs (SRS13-45 and SRS13-60) significantly inhibited glutamate release, while two non-lethal analogs (EC$_{50}$ >40 μM), SRS13-67 and SRS14-98, did not (*Figure 6C*). The ability of these analogs to induce caspase-3/7 (DEVDase) activity did not vary significantly from one another or from DMSO-treated controls, as compared to cells treated with a positive control inducer of apoptosis, the proteasome inhibitor MG132 (*Figure 6D*). Together, these results may help account for other reports of caspase-independent, sorafenib-induced death (*Panka et al., 2006*; *Katz et al., 2009*) and support the hypothesis that sorafenib triggers ferroptotic cell death via, possibly indirect, inhibition of system $x_c^-$.

## Association of sorafenib with a unique constellation of adverse clinical events

Sorafenib is a clinically-used drug for the treatment of renal cell carcinoma and other indications. We speculated that the ability of sorafenib to trigger both ferroptotic and non-ferroptotic death would result in a unique spectrum of clinical observations in patients treated with sorafenib compared to other kinase inhibitors. Specifically, a subset of patients for any drug typically experience adverse events that are dependent on the drug mechanism. Thus, we speculated that the pattern of adverse events could report on the similarity or differences of mechanisms across drugs. Previously, we applied a large-scale statistical analysis to the Food and Drug Administration Adverse Event Reporting System (FAERS) to systematically identify drug effects and interactions (*Tatonetti et al., 2012*). Here, we sought to use this approach to discover correlations between sorafenib exposure and human health unique to this drug. First, we identified those reports of patients with exposure to sorafenib, and a set of reports that could serve as controls; for this, we used a high dimensional propensity-score model previously validated for this use in the FAERS that has been shown to mitigate confounding bias and to improve the accuracy of statistical estimates (*Tatonetti et al., 2012*). Using disproportionality analysis (*Bate and Evans, 2009*), we identified adverse drug effects for sorafenib and for a set of comparison kinase-targeted drugs for which sufficient data was available in our data (dasatinib, erlotinib, gefitinib, imatinib, lapatinib, and sunitinib), none of which (at 20 μM) were found to trigger ferroptosis in HT-1080 cells or inhibit system $x_c^-$ activity as assayed using the glutamate release assay (*Figure 7—figure supplement 1*). We then filtered out effects that could be attributed to chemotherapy and grouped the drug-effect associations by the physiological system that the adverse event affected. For example, cardiovascular-related adverse events were grouped into the cardiovascular system category. For each kinase inhibitor, we counted the number of reports in each of 20 physiological system categories that involved the above drugs. We then compared this number to the counts obtained from the selected control cohorts. Using a Fisher's exact test, we evaluated significance of associations of each kinase inhibitor to each physiological system category, and then we plotted these data as a heatmap, clustering the data in an unsupervised, hierarchical manner using the computed p values (*Figure 7*).

In this analysis, we observed that sorafenib treatment was associated with a significant number of adverse events in 15/20 physiological system categories, the most observed for any drug. A subset of the adverse events uniquely associated with sorafenib compared to all other kinase inhibitor drugs included musculoskeletal, nervous system, and pathological disorders as well as hemorrhage; this pattern was not observed for patients treated with sunitinib, which is approved for the same indication as sorafenib (*Stein and Flaherty, 2007*), making it unlikely that these events are confounded by the

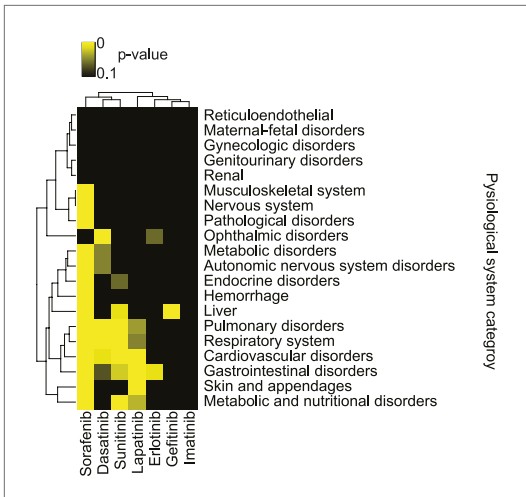

**Figure 7**. Summary of adverse events reported with sorafenib and other kinase inhibitors. Analysis of adverse events across 20 physiological system categories associated with kinase inhibitor treatment.

The following figure supplements are available for figure 7:

**Figure supplement 1**. Test of the ability of various kinase inhibitors to induce ferroptosis and inhibit system $x_c^-$.

particular patient population under study. These results suggest that, compared to other clinically-approved kinase inhibitors, sorafenib treatment has an increased propensity to generate unexpected adverse events in a variety of physiological systems. While these data are merely correlative at this point, one possibility, given the unique ability of sorafenib to inhibit system $x_c^-$ among tested kinase inhibitors, is that this effect contributes to increase the chance of an adverse event in combination with one or more underlying modifying factors.

## Upregulation of *AKR1C* family genes is associated with resistance to ferroptosis

Our results suggested that inhibition of system $x_c^-$ function by erastin, SAS and sorafenib (within a narrow concentration range) triggers ferroptosis. We sought to identify genetic modifiers of this process in erastin-resistant cell lines. First, we isolated from DU-145 prostate cancer cells five clonal cell lines that displayed significant (>three-fold) resistance to the lethal effects of erastin but not the multidrug resistance pump substrate Taxol (paclitaxel) (*Figure 8A–C*). These five resistant cell lines displayed significant resistance to additional ferroptosis inducers including SAS, sorafenib and the more potent erastin analogs identified above (**20**–**22**) (*Figure 8D,E*, *Figure 8—figure supplement 1*). These lines also displayed resistance to RSL3, which triggers ferroptosis not through inhibition of system $x_c^-$ but through the inhibition of the glutathione peroxidase GPX4 (*Yang et al., 2014*; *Figure 8—figure supplement 1*). This observation suggested that resistance was unlikely to be due to any effect on upstream cystine import or glutathione production. Indeed, using the glutamate release assay, we found that system $x_c^-$ was equally sensitive to the inhibitory effects of erastin in the parental DU-145 line and the five resistant cell lines (*Figure 8F*). Likewise, the parental and resistant cell lines exhibited largely equivalent levels of basal total glutathione and depletion of glutathione following erastin treatment (*Figure 8G*).

To explore further potential mechanisms of resistance, we examined basal and erastin-stimulated ROS levels in parental DU-145 cells and a subset of the erastin-resistant clones. We observed that the resistant cell lines exhibited substantially lower levels of basal and erastin-induced ROS accumulation, as detected by $H_2DCFDA$ using flow cytometry (*Figure 8H*). This result suggested that resistance to various ferroptosis inducers was likely due to an inhibition of the accumulation of lethal oxidative species.

To identify candidate genes involved in this process, we used RNA Seq to identify changes in gene expression associated with resistance. In this analysis, we focused on genes that were transcriptionally upregulated in resistant clones vs the parental cell line. In total, we identified 73 genes that were upregulated at least 10-fold on average across the five resistant cell lines compared to the parental clones (Data available as 'Data Package 2' from Dryad data Repository [*Dixon et al., 2014*]). The two genes that exhibited the highest average fold-change in expression and that were upregulated to highest average absolute levels within the cell (e.g., FPKM >100) were *AKR1C1* (586-fold up-regulation) and *AKR1C2* (528-fold up-regulation) (*Figure 8I*). A third family member, *AKR1C3* was upregulated 84-fold. The AKR1C1-3 enzymes have been shown to participate in the detoxification of toxic lipid metabolites (such as 4-hydroxynonenal) generated downsteam of the oxidation of various polyunsaturated fatty acid species (*Figure 8J*; *Burczynski et al., 2001*). Thus, overexpression of *AKR1C* family members (and potentially other genes) may confer partial resistance to erastin by enhancing the detoxification of reactive aldehydes generated downstream of the oxidative destruction of the plasma membrane during ferroptosis.

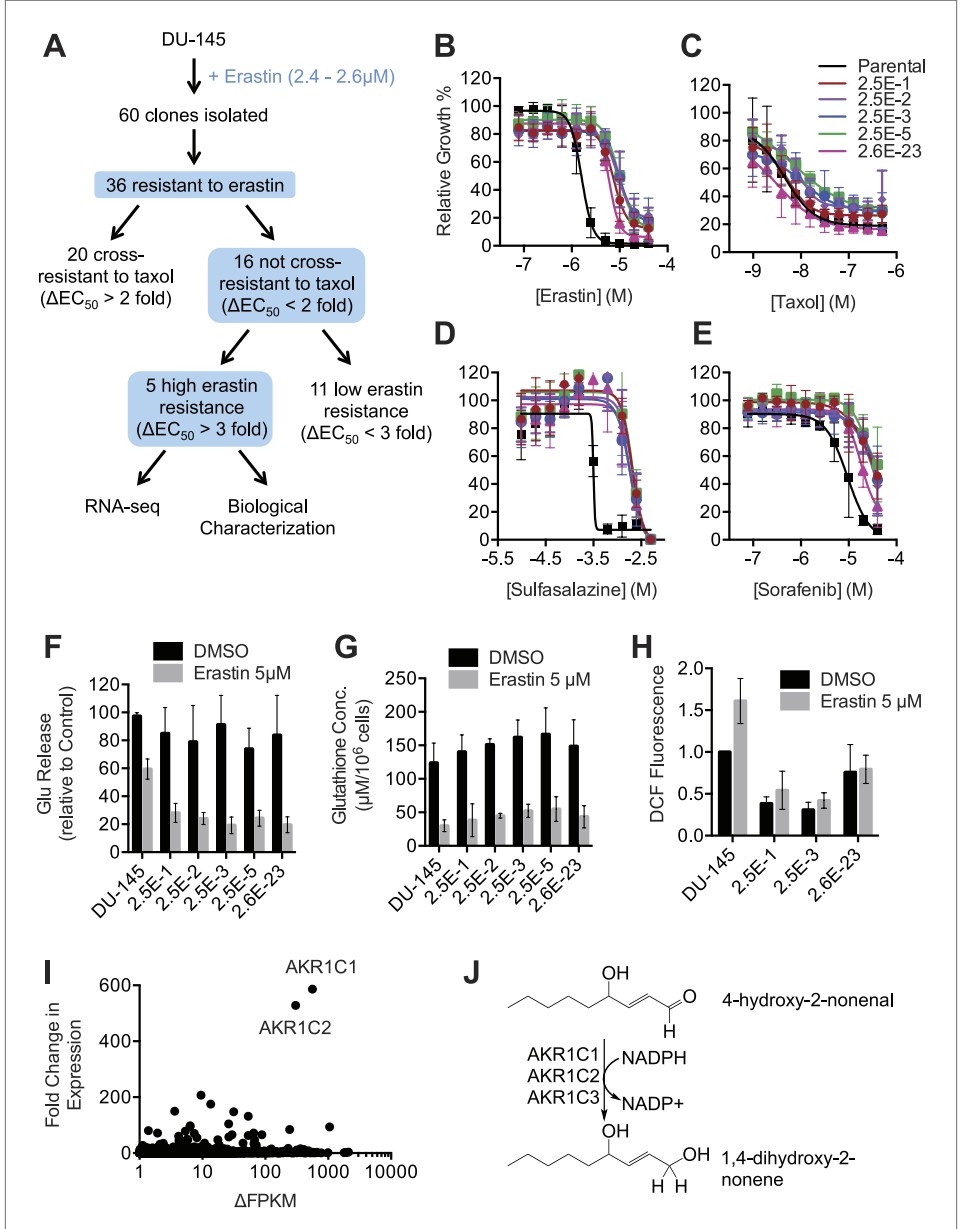

**Figure 8**. Isolation and analysis of erastin-resistant clones identifies *AKR1C* genes as mediators of resistance to system $x_c^-$ inhibition. (**A**) Outline of the isolation of DU-145 erastin-resistant clones. (**B**–**H**) Comparison of the parental DU-145 cell line and five erastin-resistant clonal lines to the indicated lethal compounds: (**B**–**E**), response to different lethal compounds, (**F**) Glutamate release, (**G**) Glutathione concentration, (**H**) DCF fluorescence. (**I**) Summary of RNA Seq analysis of the five erastin-resistant clones vs the parental DU-145 cells. Fold-change in expression (y-axis) and absolute change in FPKM (x-axis) is computed from the average of the five resistant cells lines vs parental. (**J**) Summary of AKR1C family member activity possibly relevant to ferroptosis.

The following figure supplements are available for figure 8:

**Figure supplement 1**. Test of the ability of additional ferroptosis inducers to cause death in parental DU-145 and erastin-resistant DU-145 cell lines.

## Discussion

Amino acid transporters, such as system $x_c^-$, are potentially attractive drug targets, as these proteins are crucial for cell survival and growth. In this study, we have demonstrated that erastin is a potent and

specific inhibitor of system $x_c^-$-mediated cystine uptake and further elucidated the mechanism of action of this and other system $x_c^-$ inhibitors that are able to trigger ferroptosis. Previous metabolomics analysis had suggested that erastin inhibited system L (SLC3A2 + SLC7A5)-mediated amino acid transport in Jurkat T cells (*Dixon et al., 2012*). It was therefore surprising to observe that in HT-1080 and Calu-1 cells erastin inhibited system $x_c^-$ (SLC3A2 + SLC7A11)-mediated cystine uptake, but not system-L-mediated phenylalanine uptake. These results rule out the possibility that erastin inhibits SLC3A2-dependent transporters non-specifically. Further, given evidence that Jurkat cells do not express system $x_c^-$ (*Kakazu et al., 2011*), we hypothesize that in cells lacking system $x_c^-$, erastin can inhibit structurally-related transporters (e.g., system L). An alternative hypothesis is that erastin binds to some indirect target that, in Jurkat cells, favors the inhibition of system L, while in HT-1080, Calu-1 and possibly other cells, favors the inhibition of system $x_c^-$. A definitive resolution of this matter will require further study.

An important goal is to identify scaffolds capable of inhibiting system $x_c^-$ with greater potency than existing compounds typified by SAS and derivatives (*Gorrini et al., 2013*). We found that erastin is a substantially more potent inhibitor of system $x_c^-$ function than SAS. Further optimization of the erastin scaffold yielded analogs with improved potency against this antiporter in the low nanomolar range that retained a degree of selectivity towards oncogenic mutant HRAS-expressing cells. Our work suggests that the ability of erastin, SAS and sorafenib to induce ferroptosis is tied to the inhibition of system $x_c^-$-mediated cystine import, and the consequent depletion of glutathione and loss of GPX4 activity (*Yang et al., 2014*). However, these compounds are also predicted to inactivate the cystine/cysteine redox cycle (*Banjac et al., 2008*) and, by restricting the intracellular supply of cysteine, to inhibit new protein synthesis. Together, these effects may further reduce cell growth or cause cell death in certain contexts where the induction of ferroptosis per se is not possible. The ability of system $x_c^-$ inhibitors such as erastin to trigger ferroptosis at similar concentrations in both monolayer and three-dimensional MCTS cultures, and at both high (21%) and low (1%) $O_2$ levels, suggests that these compounds are capable of overcoming the 'multicellular resistance' phenomenon observed with many lethal molecules (*Desoize and Jardillier, 2000*) and do not necessarily require high levels of ambient $O_2$ to be lethal.

Consistent with the hypothesis that erastin treatment deprives cells of cystine, RNA Seq expression profiling and subsequent follow-up studies of erastin-treated cells revealed transcriptional upregulation of *DDIT3* (CHOP), *DDIT4* (REDD1) and *ATF3*, canonical targets of the eIF2alpha-ATF4 branch of the unfolded protein response (UPR)/ER stress pathway, which we also showed to be activated biochemically. These results are consistent with previous work showing upregulation of these genes in mammalian cells cultured in the absence of cysteine (*Lee et al., 2008*). Intriguingly, ATF4 is thought to be an important regulator of *SLC7A11* expression and deletion of *Atf4* in mouse embryonic fibroblast cells results in an oxidative, iron-dependent death phenotype that is highly reminiscent of ferroptosis (*Harding et al., 2003*), suggesting that the basal level of ATF4 may set the threshold for ferroptotic death. *CHAC1* is a downstream target of the eIF2alpha-ATF4 pathway (*Mungrue et al., 2009*) and our results suggest that *CHAC1* upregulation may be useful as a PD marker for cystine or cysteine-starved cells. Whether CHAC1 plays a role in the execution of ferroptosis remains unclear. ChaC-family proteins were recently reported to function in yeast as intracellular glutathione-degrading enzymes (*Kumar et al., 2012*). One possibility is that CHAC1 upregulation following system $x_c^-$ inhibition may actively contribute to glutathione depletion in cells deprived of cysteine, although inhibition of *CHAC1* transcription had only minimal effects on the viability of erastin-treated cells.

The finding that sorafenib can inhibit system $x_c^-$ was unexpected. Despite sorafenib's multi-kinase inhibitory activity, there is disagreement about whether the sorafenib lethal mechanism of action in cells involves kinase inhibition or binding to an alternative target (*Wilhelm et al., 2008*). It has previously been shown that sorafenib treatment inhibits translation (*Rahmani et al., 2005*), induces ER stress and the expression of *DDIT4* (REDD1) via the eIF2alpha-ATF4 pathway (*Rahmani et al., 2007*; *Kim et al., 2011*), and causes caspase-independent death (*Panka et al., 2006*; *Katz et al., 2009*). Most recently, it was suggested that in hepatocellular carcinoma cells, sorafenib can trigger iron-dependent death (*Louandre et al., 2013*). Here, we show that inhibition of system $x_c^-$-mediated cystine import by sorafenib can lead to both the induction of an ER stress response (as indicated by phosphorylation of eIF2alpha and upregulation of both ATF4 and *CHAC1*) and ferroptotic cell death. Thus, our results provide a satisfying mechanistic explanation for previous observations: namely, that sorafenib can inhibit system $x_c^-$ function, leading to ER stress and in some cells the induction of non-apoptotic, iron-dependent, ferroptotic cell death.

Data collected in Phase I clinical trials of sorafenib-treated patients demonstrate that at clinically recommended doses (400 mg), it is possible to achieve maximum plasma concentration of sorafenib

from 5.2–21 µM (*Awada et al., 2005*; *Strumberg et al., 2005*). This encompasses the range within which we observe the ferroptosis-inducing effects of sorafenib. Notably, sera collected from sorafenib-treated patients display evidence of protein oxidation, with higher levels of protein oxidation being correlated with improved patient outcomes (*Coriat et al., 2012*). Thus, it is conceivable that sorafenib could be having effects in vivo related to the inhibition of system $x_c^-$ function and the subsequent generation of reactive oxygen species. Indeed, given that renal cell carcinomas are among the most sensitive of all cancer cell lines to the lethal effects of erastin (*Yang et al., 2014*), it may be of interest to re-evaluate whether the efficacy of sorafenib observed in patients could be due, at least in part, to inhibition of system $x_c^-$-mediated cystine uptake. Likewise, adverse events that are observed in a minority of patients treated with sorafenib may be due to inhibition of system $x_c^-$. While not all patients treated with sorafenib experience adverse events, we suspect that specific underlying (and unknown) sensitizing factors will render a minority of individuals more sensitive to these events. It would be important to account for this potential toxicity in the design of future therapeutics.

As with many molecularly targeted compounds (*Holohan et al., 2013*), the ultimate clinical utility of system $x_c^-$ inhibition will be influenced by the ability of target cells to evolve resistance to these inhibitors. We found that exposure to erastin can result in the emergence of cell populations partially resistant to this compound due to dramatic overexpression of multiple *AKR1C* family members. These enzymes have a number of substrates but have been shown to detoxify toxic lipid metabolites such as 4-HNE (*Burczynski et al., 2001*), which are likely produced by the oxidative lipid fragmentation processes that occur during the execution of ferroptosis (*Dixon et al., 2012*; *Skouta et al., 2014*; *Yang et al., 2014*). The expression of the AKR1C genes is controlled by the antioxidant master regulatory transcription factor NRF2, which itself is under the negative regulation of KEAP1 (*Lou et al., 2006*; *Agyeman et al., 2012*; *Jung and Kwak, 2013*). Mutations of both these genes are observed in numerous cancers (*Jaramillo and Zhang, 2013*), and we would predict that these changes enhance AKR1C expression and possibly render cells resistant to the induction of ferroptosis downstream of system $x_c^-$ inhibition.

## Materials and methods

### Chemicals
Erastin was synthesized as described (*Yagoda et al., 2007*). Additional erastin and sorafenib analogs were prepared as described in the *Supplementary file 1*. Data also available as 'Extended Materials and Methods' from Dryad data Repository (*Dixon et al., 2014*). The synthesis of (1*S*, 3*R*)-RSL3 was described (*Yang et al., 2014*). Sorafenib, imatinib, erlotinib, lapatinib, nilotenib, dasatinib, sunitinib, and gefetinib were from SelleckChem (Houston, USA). Unless otherwise indicated, all other compounds were from Sigma-Aldrich (St. Louis, USA).

### Cell lines and media
BJeH, BJeHLT and BJeLR cells were obtained from Robert Weinberg (Whitehead Institute). 143B cells were obtained from Eric Schon (Columbia University Medical Center). HT-1080 and Calu-1 cells were obtained from American Type Culture Collection. BJeH, BJeHLT and BJeLR cells were grown in DMEM High-Glucose media (Gibco/Life Technologies Corp., Grand Island, NY) plus 20% M199 (Sigma, St. Louis, MO) and 15% heat-inactivated fetal bovine serum (FBS). HT-1080 cells were grown in DMEM High-Glucose medium (Gibco) supplemented with 10% FBS and 1% non-essential amino acids (Gibco). Calu-1 and U2OS cells were grown in McCoy's 5A media (Gibco) supplemented with 10% fetal bovine serum. MEFs were grown in DMEM supplemented with 10% fetal calf serum. 143B cells were grown in DMEM High-Glucose supplemented with 10% FBS. All cell lines were grown in humidified tissue culture incubators (Thermo Scientific) at 37°C with 5% $CO_2$. Except where indicated, all media were supplemented with penicillin and streptomycin (Gibco).

### Growth in low oxygen conditions
For low oxygen experiments cells were grown under normal (21%) oxygen conditions, then split into two 6-well dishes, one of which was cultured at 21% $O_2$/5% $CO_2$ in a regular tissue culture incubator, and one of which was transferred to a HypOxygen H35 incubator for growth under 1% $O_2$/5% $CO_2$ conditions for 24 hr. The next day compounds were added directly to the plates. In the case of the 1% $O_2$, condition, compound addition was done within the confines of the chamber to prevent media re-oxygenation. 24 hr later, cells were removed from the chamber and viability was assessed immediately by Vi-Cell.

## Multicellular tumor spheroid assays

Multicellular tumor spheroids (MCTSs) were grown in 96-well Corningware Ultra Low Attachment (ULA) Plates (CLS 3474). 200 µl of cell suspension containing $10^4$ cells/ml were added to each well of the ULA plate, after which they were incubated at 37°C/5% $CO_2$ for 72 hr to allow for MCTS formation. MCTSs were then treated with lethal compounds (vehicle control [DMSO], 10 µM Erastin, 1 µM RSL3, or 1 µM STS) ±inhibitors (vehicle control [DMSO], 1 µM Ferrostatin-1, or 25 µM β-mercaptoethanol) by carefully aspirating 50 µl of media from each well, and replacing with 50 µl each of media containing 4 × desired treatment concentration of the lethal or inhibitor. After 72 hr of treatment, MCTS images were acquired using an EVOS fl microscope (Advanced Microscopy Group/Life Technologies Corp.) equipped with a 10× phase contrast objective. Three independent fields were acquired for each experimental condition. Representative samples from one field of view are shown. Viability was then measured using Alamar blue as described above and measured on a Victor3 platereader.

## Radioactive uptake assays

200,000 HT-1080 or Calu-1 cells/well were seeded overnight in 6-well dishes (Corning Life Sciences, Tewksbury, MA). The next day, cells were washed twice in pre-warmed $Na^+$-free uptake buffer (137 mM choline chloride, 3 mM KCl, 1 mM $CaCl_2$, 1 mM $MgCl_2$, 5 mM D-glucose, 0.7 mM $K_2HPO_4$, 10 mM HEPES, pH 7.4), then incubated for 10 min at 37°C in 1 ml of uptake buffer, to deplete cellular amino acids. At this point, in each well the buffer was replaced with 600 µl uptake buffer containing compound and 0.12 µCi (80–110 mCi/mmol) of L-[3,3'-$^{14}$C]-cystine or 0.2 µCi of L-[$^{14}$C(U)]-phenylalanine (PerkinElmer, Waltham, MA) and incubated for 3 min at 37°C. Cells were then washed three times with ice-cold uptake buffer and lysed in 500 µl 0.1 M NaOH. To this lysate was added 15 ml of scintillation fluid, and radioactive counts per minute were obtained using a scintillation counter. All experiments were repeated in three independent biological replicates for each condition. To control for differences in the absolute counts of radioactivity between replicates, data were first normalized to DMSO (set to 100%) within each replicate, then averaged across three biological replicates.

## Medium-throughput glutamate release assay

The release of glutamate from HT-1080 cells into the extracellular medium was detected using an Amplex Red glutamate release assay kit (Molecular Probes/Life Technologies Corp., Eugene, OR). For compound treatment experiments, 200,000 cells/well were seeded overnight into 6-well dishes (Corning). The next day, cells were washed twice in PBS and then incubated for one hour in $Na^+$-containing, glutamine-free media (Cellgro/Corning) containing various compounds at different concentrations. For siRNA experiments, cells were transfected with siRNAs for 48 hr (see above), then washed twice in PBS and incubated for an hour in $Na^+$-containing, glutamine-free media. 50 µl of medium per well was removed and transferred to a 96-well assay plate (Corning) and incubated with 50 µl of a reaction mixture containing glutamate oxidase, L-alanine, glutamate-pyruvate transaminase, horseradish peroxidase, and Amplex Red reagents as per the manufacturer's protocol. Glutamate release was first normalized to total cell number determined by Vi-Cell counting at the end of the experiment, then values were expressed as a percentage of no treatment (DMSO) controls. In some experiments, a glutamate standard curve was used to quantify the exact amount of glutamate release. Of note: as the medium contained $Na^+$, the total amount of glutamate release reflects the activity of both system $x_c^-$ ($Na^+$-independent) and non-system $x_c^-$ glutamate transporters and therefore never reaches 100% inhibition, as system $x_c^-$ accounts for only a portion of total glutamate release.

## High-throughput glutamate release assay

This glutamate release assay was used during the testing of erastin analogs. Human astrocytoma cells (CCF-STTG1) were used to assay cystine–glutamate antiporter ($x_c^-$) activity. Cells, cultured in RPMI + 10% FBS, were grown in 96-well plates; when confluent, cells were washed with Earle's Balanced Salt Solution (EBSS, Sigma) containing $Ca^{2+}$ and $Mg^{2+}$ to remove residual glutamate. Cells were then incubated for 2 hr at 37°C with either EBSS/glucose (blanks) or EBSS/glucose containing 80 µM cystine (totals) ± compounds (30 nM–100 µM). The known $x_c^-$ inhibitor, (S)-4-carboxyphenylglycine (S-4CPG), was used as positive control. Following incubation, glutamate released into medium was detected using the Amplex Red system (Life Technologies), as per the manufacturer's instructions.

## siRNA reverse transfection

HT-1080 cells were reverse transfected with siRNAs (Qiagen, Germantown, MD) using Lipofectamine RNAiMAX (LFMax, Invitrogen/Life Technologies Corp.). Briefly, 1–10 nM (final concentration) of

siRNAs was aliquoted into 250 µl Opti-MEM media (Gibco) in the bottom of each well of a 6-well dish (Corning). An additional 250 µl medium + LFMax was added to each well and incubated for 15 min. At this point,150,000 HT-1080 cells were added to each well in regular HT-1080 medium. The plates were swirled to mix and incubated for 48 hr at 37°C in a tissue culture incubator prior to analysis.

## Reverse transcription-quantitative polymerase chain reaction (RT-qPCR)

RNA was extracted using the Qiashredder and Qiagen RNeasy Mini kits (Qiagen) according to the manufacturer's protocol. 1–2 µg total RNA for each sample was used as input for each reverse transcription reaction, performed using the TaqMan RT kit (Applied Biosystems/Life Technologies Corp., Foster City, CA). Primer pairs for were designed for target transcripts using Primer Express 2.0 (Applied Biosystems). Quantitative PCR reactions were performed using the Power SYBR Green PCR Master Mix (Applied Biosystems). Triplicate samples per condition were analyzed on an Applied Biosystems StepOnePlus qPCR instrument using absolute quantification settings. Differences in mRNA levels compared to *ACTB* internal reference control were computed between control and experimental conditions using the ΔΔCt method.

## Cell viability measurements

Cell viability was typically assessed in 384-well format by Alamar Blue (Invitrogen, Carlsbad, CA) fluorescence (ex/em 530/590) measured on a Victor3 platereader (PerkinElmer). In some experiments, Trypan Blue dye exclusion counting was performed using an automated cell counter (ViCell, Beckman–Coulter, Fullerton, CA). Cell viability in test conditions is reported as a percentage relative to the negative control treatment.

## Glutathione level assay

Total intracellular glutathione (GSH+GSSG) was measured using a glutathione assay kit based on Ellman's reagent (Cayman Chemical #703002; Ann Arbor, USA) according to instructions. 200,000 HT-1080 cells per well were seeded overnight in 6-well dishes (Corning). The next day, cells were treated with compounds (erastin for 5 hr, sorafenib for 18 hr), then washed once in 500 µl PBS and harvested by scraping into phosphate buffer (10 mM X, 1 mM EGTA). Cells were then lysed by sonication (7 cycles, 2 s on, 1 s off) and spun at 4°C for 15 min at 13,000 rpm to pellet membranes. Supernatants were mixed with 500 µl of a 10% solution of metaphosphoric acid (wt/vol), vortexed briefly, and centrifuged for 3 min at 4000 rpm. The supernatant was transferred to a new tube and to this was added 50 µl of triethanolamine solution (4 M). This was vortexed and 50 µl per sample was aliquoted to each well of a 96-well plate. 150 µl of assay buffer containing 5,5'-dithiobis(2-nitrobenzoic acid) (DTNB, Ellman's reagent) was added to each well and the reaction was incubated for 25 min at room temperature with rotation, at which point absorbance was measured at 405 nM. The glutathione concentration was calculated in reference to a glutathione standard curve and normalized to total cell number per well, as determined from parallel plates.

## Flow cytometry

Flow cytometry was performed using an Accuri C6 flow cytometer equipped with a 488 nm laser. Reactive oxygen species accumulation was assessed using $H_2DCFDA$ and C11-BODIPY 581/591 (both from Molecular Probes/Life Technologies) as described in *Dixon et al. (2012)*.

## RNA-sequencing

RNA was isolated from compound-treated HT-1080 cells, or from DU-145 parental and erastin-resistant cell lines, as described for RT-qPCR reactions. Poly-A pull-down was then used to enrich mRNAs from total RNA samples (1 µg per sample, RIN >8) and libraries were prepared using Illumina TruSeq RNA prep kit (San Diego, CA). Libraries were then sequenced using an Illumina HiSeq 2000 instrument (Columbia Genome Center, Columbia University). For the analysis of gene upregulation by erastin treatment, five samples were multiplexed in each lane, to yield the targeted number of single-end 100 bp reads for each sample (30 million), as a fraction of 180 million total reads for the whole lane. For the analysis of erastin-resistant cell lines, paired-end 100 bp reads for each sample (60 million) were collected. Short reads were mapped to the human reference genome using Tophat (*Trapnell et al., 2009*). The relative abundance of genes and splice isoforms was determined using Cufflinks (*Trapnell et al., 2010*). We then looked for differentially expressed genes under the various experimental conditions using Cuffdiff, a program included in the Cufflinks package. For the HT-1080 experiments, only genes with FPKM not equal to zero in any condition and FPKM ≥1 for both replicates in the DMSO-treated condition were considered in the analysis. To further restrict the analysis to high quality data, we only examined those

genes where the difference between replicate values in the two DMSO and the two erastin-treated samples was <2.5x.

## Modulatory effect profiling

Modulatory effect ($M_e$) profiling was performed as described (*Wolpaw et al., 2011*; *Dixon et al., 2012*). In *Figure 1A,B*, the following ferroptosis inhibitors were tested (maximum concentration in 10 point, twofold dilution series listed here): cycloheximide (CHX, 50 µM), ferrostatin-1 (Fer-1, 2 µM), trolox (300 µM), U0126 (15 µM), ciclopirox olamine (CPX, 50 µM), and beta-mercaptoethanol (β-ME, 20 µM). In *Figure 5A,B*, the following lethal compounds were tested (maximum concentration in 10 point, twofold dilution series listed here): methotrexate (100 µM), bortezomib (5 µM), doxorubicin (50 µM), chlorambucil (500 µM), irinotecan (5 µM), SAHA (50 µM), actinomycin D (1 µg/ml), gefitinib (50 µM), taxol (5 µM), sorafenib (10 µM), erlotinib (5 µM), GX15 (10 µM), staurosporine (STS, 2 µM), phenylarsine oxide (PAO, 1 µM), RSL3 (5 µM), $H_2O_2$ (5 mM), ABT-263 (50 µM), 6-thioguanine (5 µM), 17-AAG (5 µM) and imatinib (50 µM).

## Western blotting

Cells were lysed in lysis buffer consisting of 50 mM HEPES, 40 mM NaCl, 2 nM EDTA, 0.5% Triton X-100, 1.5 mM $Na_3VO_4$, 50 mM NaF, 10 mM sodium pyrophosphate, 10 mM sodium beta-glycerophosphate and 1 tablet of protease inhibitor. Cell lysates were separated on a 4–20% Tris-gel, transferred to nitrocellulose membrane and blocked in PBST +5% milk for an hour. Membranes were incubated with primary antibody at the following concentrations: ATF4 (sc-200; Santa Cruz, Santa Cruz, CA) 1:100, phosphor-eIF2α (5324; Cell Signaling, Danvers, MA) 1:200, eIF2α (3597; Cell Signaling) 1:1000, p-PERK 1:200 (sc-32577; Santa Cruz), PERK 1:200 (sc-13073; Santa Cruz), BiP (3177; Cell Signaling) 1:1000, eIF4E (610270; BD Biosciences, San Jose, CA) 1:1000 in PBS + 5% BSA overnight at 4°C. Next day washed and incubated with 1:2000 HRP-conjugated secondary before development with SuperSignal West Pico Substrate (#34080; Pierce, Rockford, IL).

## XBP-1 PCR analysis

HT-1080 cells were compound treated then mRNA was harvested and 2 µg of mRNA used as a template for first-strand cDNA synthesis, also as described above for RT-qPCR. The cDNA was used as input for a PCR reaction using XBP-1-specific primers to detect splicing (Forward [5′-3′]: TTACGAGAG AAAACTCATGGCC, Reverse [5′-3′]: GGGTCCAAGTTGTCCAGAATGC) and actin B-specific primers as a control. PCR conditions were as follows: 94°C for 1 min, followed by 35 cycles of 94°C for 30 s, 60°C for 30 s, 72°C for 1 min. PCR products were separated on a 2.5% agarose gel and visualized using an Syngene G:Box imaging station.

## Caspase-3/7 activity assay

Cells were seeded (1500 cells/well) in a volume of 40 µl medium in 384-well plates (Corning) for 24 hr prior to treatment of lethals and/or inhibitors. Following compound treatment for 24 hr, 10 µl of a 1:100 vol/vol dilution of Apo-One Homogeneous Caspase 3/7 substrate solution/assay buffer (Promega, Madison, WI) was added to samples, and the plate was vigorously agitated for 30 s. Plates were then incubated for 8 hr in the dark at room temperature, allowing for caspase-3/7 cleavage of the fluorogenic substrate, before measuring fluorescence at excitation/emission wavelengths of 498/521 nm using a Victor3 plate reader (Perkin Elmer).

## Biological data collection and statistical analyses

Except where indicated, all experiments were performed at least three times on separate days as independent biological replicates. The data shown represents the mean ± SD of these replicates. All statistical analyses and curve fitting were performed using Prism 5.0c (GraphPad Software, La Jolla, CA).

## Bioinformatics

Gene Ontology (GO) process enrichment was computed using the web-based GOrilla tool with default settings (*Eden et al., 2009*; http://cbl-gorilla.cs.technion.ac.il).

## Clinical data description

The Food and Drug Administration (FDA) collects and maintains spontaneously submitted adverse event reports in the Adverse Event Reporting System (FAERS). We downloaded 2.9 million adverse

events from FAERS representing reports through the fourth quarter of 2011. To complement FAERS, we also downloaded the OFFSIDES drug effect database (*Tatonetti et al., 2012*). In addition, we extracted the laboratory values, clinical notes, prescription orders, and diagnosis billing codes from the electronic health records (EHR) at Columbia University Medical Center/New York Presbyterian Hospital for 316 patients with at least one prescription order of sorafenib, dasatinib, erlotinib, gefetinib, imatinib, lapatinib, or sunitunib. These data were employed in the analysis (described in detail in the main text) to identify adverse events uniquely associated with sorafenib treatment. This analysis was covered under the Columbia Institutional Review Board (IRB) protocol number AAAL0601.

### Isolation of erastin-resistant clones

To generate resistant clones, DU-145 prostate cancer cells were seeded in 10-cm dishes with complete medium containing ~3 × $EC_{50}$ erastin (2.4 μM or 2.6 μM), which was found to be effective at initially reducing the population on any given plate to a small number of individual surviving cells. The erastin-supplemented medium was replaced every 3 days for 2–3 weeks to allow for clonal expansion. As summarized in *Figure 7A*, 60 resistant clones were initially isolated by ring cloning, with selection limited to non-diffuse cell clusters to minimize risk of selecting drifted populations. Clones were subsequently maintained in complete medium without erastin. Of 36 clones found to be resistant to erastin in an initial re-testing, 20 were cross-resistant to Taxol (paclitaxel), a mechanistically unrelated drug. Of the remaining 16 clones, five exhibited strong resistance to erastin, but not to Taxol, and were further characterized.

## Acknowledgements

We gratefully acknowledge the assistance of Dr John Decatur, and the use of Columbia Chemistry NMR core facility instruments provided by NSF grant CHE 0840451 and NIH grant 1S10RR025431-01A1. Some small molecule synthesis was performed by Rachid Skouta, Elise Jiang and Reed Northland in the Columbia NYSTEM Chemical Probe Synthesis Facility (Contract No. C026715).

## Additional information

### Funding

| Funder | Grant reference number | Author |
|---|---|---|
| Howard Hughes Medical Institute | Brent Stockwell Award | Brent R Stockwell |
| National Institutes of Health | 5R01CA097061, 5R01GM085081 and R01CA161061 | Brent R Stockwell |
| New York Stem Cell Science | C026715 | Brent R Stockwell |
| National Institutes of Health | 1K99CA166517-01 | Scott J Dixon |

The funders had no role in study design, data collection and interpretation, or the decision to submit the work for publication.

### Author contributions

SJD, DNP, MW, Conception and design, Acquisition of data, Analysis and interpretation of data, Drafting or revising the article; RS, Conception and design, Acquisition of data, Analysis and interpretation of data, Drafting or revising the article, Contributed unpublished essential data or reagents; EDL, MH, AGT, CEG, NPT, BSS, Acquisition of data, Analysis and interpretation of data, Drafting or revising the article; BRS, Conception and design, Analysis and interpretation of data, Drafting or revising the article

### Author ORCIDs

Brent R Stockwell, http://orcid.org/0000-0002-3532-3868

### Ethics

Human subjects: The data analysis described in this manuscript was covered under the Columbia Institutional Review Board (IRB) protocol number AAAL0601 and performed according to NIH and Columbia University guidelines.

# Additional files

## Supplementary file

• Supplementary file 1. Extended materials and methods. Description of chemical synthesis and characterization. Data also available as 'Extended Materials and Methods' from Dryad data Repository: http://dx.doi.org/10.5061/dryad.jp43c.

## Major datasets

The following datasets were generated:

| Author(s) | Year | Dataset title | Dataset ID and/or URL | Database, license, and accessibility information |
|---|---|---|---|---|
| Scott Dixon, Darpan Patel, Matthew Welsch, Rachid Skouta, Eric Lee, Ajit G Thomas, Caroline Gleason, Nicholas Tatonetti, Barbara S Slusher, Brent R Stockwell | 2014 | Data Package 1 | doi:10.5061/dryad.jp43c/1 | Available at Dryad Digital Repository under a CC0 Public Domain Dedication. |
| Scott Dixon, Darpan Patel, Matthew Welsch, Rachid Skouta, Eric Lee, Ajit G Thomas, Caroline Gleason, Nicholas Tatonetti, Barbara S Slusher, Brent R Stockwell | 2014 | Data Package 2 | doi: 10.5061/dryad.jp43c/2 | Available at Dryad Digital Repository under a CC0 Public Domain Dedication. |
| Scott Dixon, Darpan Patel, Matthew Welsch, Rachid Skouta, Eric Lee, Ajit G Thomas, Caroline Gleason, Nicholas Tatonetti, Barbara S Slusher, Brent R Stockwell | 2014 | Extended Materials and Methods | doi:10.5061/dryad.jp43c/3 | Available at Dryad Digital Repository under a CC0 Public Domain Dedication. |

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
