## [Decision Letter]

Thank you for sending your work entitled “Pharmacological inhibition of cystine-glutamate exchange induces endoplasmic reticulum stress and ferroptosis” for consideration at *eLife*. Your article has been evaluated by a Senior editor, a Reviewing editor, and 2 reviewers. They found many of your findings to be novel and important, and recommended potential publication. However, to place your new findings in perspective with regard to prior studies, you are requested to address a number of questions.

The study explores the action of erastin, a small molecule that causes ferroptotic cell death. The key findings include 1) RNA seq analysis following erastin treatment indicates upregulation of ER stress response genes like CHAC1, ATF4, and DDIT3 2) chemical optimization of the erastin structure results in more potent antiporter x_c_^-^ inhibition, 3) data that suggest that the clinical agent sorafenib also inhibits x_c_^-^ function, and 4) clinical database analysis that suggests that x_c_^-^ inhibition may have a particular toxicological profile. The observation that erastin inhibits x_c_^-^ was previously reported by the authors in Cell in 2012. Part of the current study corroborates this previous finding with additional data; while solid and valuable, this part of the study is more confirmatory and has less impact.

The more novel aspects of the work are the identification of ER stress as a potential marker for ferroptosis, which is distinct from apoptosis and other more well characterized forms of cell death, and the possibility that sorafenib may also inhibit system x- and that this inhibition may be responsible for or contribute to observed adverse clinical effects. These more novel and higher impact aspects of the study are not investigated to the same depth, in part because the manuscript covers so many different aspects of erastin and sorafenib. As a result, the study raises a number of questions that should be addressed. In addition, the various parts of the manuscript are not as well connected as they could be (e.g., the discovery of the new erastin analogs with significantly higher potency in Figure 3 are not used in any subsequent experiments). The reviewers request you to address the questions listed below in your manuscript, either by new data or just by discussing them:

1) The study would have higher impact if a deeper mechanistic connection between the induction of ER stress genes and other features of ER stress could be explored. The demonstrated upregulation of CHAC1 is a minimal step. The authors should provide additional data that ER stress is indeed induced in ferroptosis. For instance, a correlation between the described upregulation of ER stress markers, as observed in tumor cell lines in response to erastin/sorafenib treatment, and in vivo xenotransplantation tumor models that the authors reported in Yang 2014 would strengthen the conclusion that certain ER stress genes may serve as biomarkers for pharmacodynamic investigations.

2) In Figure 4, the authors should check for statistical significance of upregulation of CHAC1 by erastin vs. STS – if there is no difference then this read-out may not be a clear discriminator.

3) With respect to sorafenib, the findings that a clinically used kinase inhibitor could function through x_c_^-^ inhibition, either directly or indirectly, is critical since the clinical community may be alarmed by this report. However, the high concentrations used in this study, 10 and 20µM, may be beyond the blood concentrations reached in patients. The PK profile of sorafenib should be compared to the in vitro activity. Also, if sorafenib is significantly protein bound, the assays in Figure 5 should be repeated in the presence of normal human serum.

4) Figures 5 and 6: Sorafenib treatment causes cell death that can be rescued by ß-ME, ferrostatin-1 and DFO, suggesting a similar mode of action as erastin. Although the authors show that glutamate secretion is inhibited by sorafenib, a direct measurement of cystine uptake, which in turn leads to glutathione depletion and lipid peroxidation, would strengthen their conclusion.

5) The structure activity relationship of the sorafenib analogs is very useful for distinguishing the relevant target. However, the authors need to measure the kinase inhibitory activity of the molecules to see if one of the active analogs has actually lost kinase binding.

6) Figure 8 provides a highly interesting peace of information that was, at present, left mostly unexplored. Erastin resistant cell lines were shown to strongly up-regulate the AKR1C enzyme family. This family of enzyme is known to be partially responsible for the detoxification of reactive aldehyde like 4-HNE. Therefore, a few questions arise: Are the resistant clones still able to undergo ferroptosis in response to different inducers? Does modulation of AKR1C expression by RNAi re-establish sensitivity to erastin?

7) In previous work (Yagoda 2007), the authors showed that erastin bound to SLC7A5, a subunit of system L, and they mention in Dixon 2012 that treatment with erastin resulted in highly significant decreases in the levels of system L substrates. These findings appear to contrast to the observations in the current work that erastin is selective for system x- and did not affect system L. The authors should discuss these apparently contradictory results in the text.

8) The authors made 19 analogs including several with significantly improved potency. They do not show the selectivity of these compounds for RAS mutant cells, which would be important for therapeutic use. A previous study from this group did report selectivity for a smaller set of analogs (Yang 2014). The selectivity should be investigated for the best analogs.

9) A statistical analysis (Figure 7) reveals sorafenib to cause multiple side effects in patients that might be partially linked to System x_c_^-^ inhibition beyond its well-documented kinase inhibitory effects. As these are mere statistical data, the statement that sorafenib indeed causes these side effects in patients partially due to system xc- inhibition is over-interpreted and should be de-emphasized. In that sense, the authors should consider that mice lacking system x_c_^-^ are fully viable and only show subtle phenotypes (Sato et al., JBC 2005). Therefore, major adverse effects induced by system x_c_^-^ inhibition in “normal” cells are less likely, which questions the validity of the statement on page 22, second paragraph, last sentence. Furthermore, the paper by [11] (referred to on page 27) describes that xCT knockout cells are in fact more resistant to against limbic seizures, and 6-OH dopamine lesioning in substantia nigra (Massie et al., FASEB 2011).

---

## [Author Response]

*1) The study would have higher impact if a deeper mechanistic connection between the induction of ER stress genes and other features of ER stress could be explored. The demonstrated upregulation of CHAC1 is a minimal step. The authors should provide additional data that ER stress is indeed induced in ferroptosis. For instance, a correlation between the described upregulation of ER stress markers, as observed in tumor cell lines in response to erastin/sorafenib treatment, and in vivo xenotransplantation tumor models that the authors reported in Yang 2014 would strengthen the conclusion that certain ER stress genes may serve as biomarkers for pharmacodynamic investigations*.

We appreciate the suggestion to further expand our studies on ER stress in the manuscript. In response, we have taken a multi-pronged approach, including new computational and experimental data to address this comment. First, we clarified that the genes upregulated by erastin are in the eIF2alpha-ATF4 branch of the ER stress response pathway, and in the updated text we now state explicitly:

“We noted that several of the genes upregulated by erastin were associated with activation of the eIF2alpha-ATF4 branch of the ER stress response pathway (e.g. ATF3, DDIT3, DDIT4 (29, 59)).”

Secondly, we analyzed our RNA-Seq transcriptome data and found that the 33 gene ‘upregulated gene’ signature is significantly enriched for GO terms related to ER stress and the UPR, supporting our hypothesis that this process is activated in erastin-treated cells. This analysis is now included as part of the updated text.

Thirdly, we now report results showing that erastin and sorafenib treatment results in upregulation of canonical biochemical markers of the eIF2alpha-ATF4 pathway, including phosphorylation of eIF2alpha and upregulation of ATF4 at the protein level. Consistent with the activation being specific to the eIF2alpha-ATF4 branch of UPR signaling, we observed no evidence for enhanced splicing of XBP-1, which is in a parallel branch of the UPR. These results are now included as part of a new Figure 4—figure supplement 1.

With respect to *CHAC1* in particular, we have made it clearer in the text that *CHAC1* expression is known to be regulated downstream of the eIF2alpha-ATF4 pathway (e.g. [40] J Immunol), which we have now shown to be activated by erastin (and sorafenib) treatment.

We also report additional data in support of our hypothesis linking system x_c_^-^ inhibition by erastin to upregulation of *CHAC1* expression. We show that, like the parent compound, several of the new erastin analogs reported here and elsewhere ([66] Cell) upregulate *CHAC1* mRNA levels, while less potently lethal erastin analogs result in more modest effects on *CHAC1* expression (new Figure 4—figure supplement 1).

In response to the request that we perform an in vivo analysis of the efficacy of *CHAC1* as a PD marker, we respectfully submit that, while these will certainly be valuable studies to perform, this is a major undertaking that is beyond the scope of what can be accomplished in the time period available to respond to these comments. We would hope to undertake these studies in the coming years as we pursue the development of these compounds in a variety of mouse tumor models.

*2) In*
Figure 4*, the authors should check for statistical significance of upregulation of CHAC1 by erastin vs. STS – if there is no difference then this read-out may not be a clear discriminator*.

We thank the reviewer for raising this point, which may have been confusing to readers. As part of our original analysis of variance (ANOVA) for this dataset we observed that *CHAC1* mRNA levels were, in fact, significantly higher in the erastin versus STS treatment condition (one-way ANOVA, *P* < .01). However, we chose not to report this previously or to do so in the updated version as we are not looking for a marker to discriminate between cells exposed to erastin versus STS, but rather are searching for a marker that might be useful to indicate exposure to erastin. As long as *CHAC1* is significantly up-regulated by erastin compared to control treatments, even if this is also observed with other unrelated compounds, *CHAC1* upregulation would still have value for in vivo studies where erastin is the only agent being administered at any one time; thus even though this comparison is statistically distinct, we respectfully suggest it would be confusing to readers and would distract from our main point to highlight it as such.

*3) With respect to sorafenib, the findings that a clinically used kinase inhibitor could function through x*_*c*_^*-*^
*inhibition, either directly or indirectly, is critical since the clinical community may be alarmed by this report. However, the high concentrations used in this study, 10 and 20µM, may be beyond the blood concentrations reached in patients. The PK profile of sorafenib should be compared to the in vitro activity. Also, if sorafenib is significantly protein bound, the assays in*
Figure 5
*should be repeated in the presence of normal human serum*.

We thank the reviewer for raising this important point. Data collected in Phase I clinical trials of sorafenib (e.g., [51]; [3]) demonstrate that at clinically recommended doses (400 mg) it is possible to achieve maximum plasma concentration of sorafenib from 5.2 - 21 μM. This encompasses the range within which we observe the system x_c_^-^ inhibiting effects of sorafenib. In this vein, it appears notable that in sera collected from sorafenib-treated patients that there is evidence for enhanced levels of oxidized proteins ([9] Mol Cancer Ther), a consequence of enhanced ROS production. Together, these results support our hypothesis that sorafenib could be having effects in vivo related to the inhibition of cystine-glutamate exchange and the induction of oxidative cell death. In the Discussion section of the updated text, we now include reference to the above literature to further bolster our arguments as follows:

“Data collected in Phase I clinical trials of sorafenib-treated patients demonstrate that at clinically recommended doses (400 mg) it is possible to achieve maximum plasma concentration of sorafenib from 5.2 - 21 μM (3, 51). This encompasses the range within which we observe the ferroptosis-inducing effects of sorafenib. Notably, sera collected from sorafenib-treated patients display evidence of protein oxidation, with higher levels of protein oxidation being correlated with improved patient outcomes (9). Thus, it is conceivable that sorafenib could be having effects in vivo related to the inhibition of system x_c_^-^ function and the subsequent generation of reactive oxygen species.”

Regarding the second point, sorafenib, like many drugs, binds human serum proteins extensively (e.g., Tod et al (2011) Pharm Res, and references therein). And, as with other drugs, serum protein (e.g., albumin) binding can have myriad effects on drug distribution that are complex and challenging to predict and model. Our experiments were performed in 10% fetal bovine serum, which contains abundant albumin to approximate the situation observed in human plasma. We could find no evidence in the literature that bovine albumin and human albumin differ substantially in drug-binding properties; thus we believe the experiments reported are sufficiently relevant to the clinical situation as to warrant reporting as they are now described.

*4)*
Figures 5 and 6*: Sorafenib treatment causes cell death that can be rescued by ß-ME, ferrostatin-1 and DFO, suggesting a similar mode of action as erastin. Although the authors show that glutamate secretion is inhibited by sorafenib, a direct measurement of cystine uptake, which in turn leads to glutathione depletion and lipid peroxidation, would strengthen their conclusion*.

We thank the reviewer for allowing us to clarify this issue, and to organize our thoughts and data to make this easier for the reader to understand. In the previously submitted manuscript, we had demonstrated that (1) sorafenib inhibits glutamate release; (2) sorafenib-induced death could be inhibited by β-ME co-treatment, which implies that death is due to the blockade of cystine uptake; and (3) sorafenib upregulates *CHAC1* expression, a feature associated with the ER stress response observed upon system x_c_^-^ inhibition. In all of these respects, sorafenib behaved like erastin.

Unfortunately, the radiolabelled cystine uptake assay is extremely challenging for us to perform for a variety of technical and logistical reasons, and is no longer available to us at the present time. However, to address this Reviewer comment, we have now validated that we are seeing the same downstream effects with sorafenib observed previously with erastin, including glutathione depletion ([66] Cell) and lipid peroxidation ([13] Cell). These new data, included in the updated Figure 5 and referenced appropriately in the updated text, show that sorafenib treatment (10 μM) results in the depletion of glutathione and the accumulation of lipid ROS, both hallmarks of cystine depletion induced by erastin. Finally, as described above, we have also tested the effects of sorafenib on biochemical markers of ER stress and shown that in all respects, sorafenib behaves like erastin. These data are now included as part of the new Figure 4—figure supplement 1. Together these results strengthen our previous findings that sorafenib inhibits system x_c_^-^ and leads to oxidative ferroptotic cell death in a manner similar to erastin.

*5) The structure activity relationship of the sorafenib analogs is very useful for distinguishing the relevant target. However, the authors need to measure the kinase inhibitory activity of the molecules to see if one of the active analogs has actually lost kinase binding*.

We appreciate the reviewers’ comment here, as we have considered approaches to addressing this issue for some time. To recap, our current working model is that sorafenib is, in fact, most likely to inhibit system x_c_^-^ activity through modulation of a kinase or kinase-like target. We do not believe that the active analogs have lost kinase binding. Rather, we argue that binding to a kinase or kinase-like target may be essential for the observed activity. Consequently, we would expect all active analogs to retain the ability to inhibit kinase activity. To help clarify this point we have re-written this section of the Results as follows:

“Several of the analogs that we synthesized were unable to trigger death (four example compounds are shown in Figure 6). These analogs (SRS13-67, SRS14-98, SRS13-48 and SRS15-11) contain modifications predicted to disrupt atomic interactions essential for the binding of sorafenib to kinase targets such as BRAF, including burial of the -CF3 group in a hydrophobic pocket and hydrogen bonding with the urea group (39, 57). Conversely, the active analogs shown here (Figure 6) mostly retain these features. Thus, one hypothesis is that sorafenib triggers ferroptosis by inhibiting an unknown kinase whose activity is necessary for constitutive system x_c_^-^ activity. Alternatively, sorafenib could modulate system x_c_^-^ activity by interacting with a non-kinase target that harbors a binding pocket resembling that found within the active site of sorafenib-sensitive kinases.”

We also recognize the reviewer’s concern for determining the source, kinase or non-kinase, for this activity. Sorafenib is known to bind to a number of kinase targets in diverse signaling pathways. We first attempted to determine the effects of active and inactive analogs on ERK phosphorylation, as it is a target of several growth signaling pathways, but we have been unable to draw definitive conclusions due to a variety of technical issues. However, analysis of effects on a particular kinase pathway would not in any case indicate whether other kinase binding activity is affected by various analogs. Comprehensive profiling across the kinome might be needed to determine whether kinase-binding activity has been altered in the active analogs. It is also possible that system x_c_^-^ inhibition by sorafenib is mediated by binding to an unknown or uncharacterized kinase, or a non-kinase target, neither of which would be resolved by a kinome-wide profiling experiment. We believe that resolving this issue requires substantial work and lies beyond the scope of this paper, which simply presents system x_c_^-^ inhibitory activity of sorafenib, and the relevant moieties of its structural scaffold necessary for this activity.

*6)*
Figure 8
*provides a highly interesting peace of information that was, at present, left mostly unexplored. Erastin resistant cell lines were shown to strongly up-regulate the AKR1C enzyme family. This family of enzyme is known to be partially responsible for the detoxification of reactive aldehyde like 4-HNE. Therefore, a few questions arise: Are the resistant clones still able to undergo ferroptosis in response to different inducers? Does modulation of AKR1C expression by RNAi re-establish sensitivity to erastin*?

The reviewers have raised important questions regarding further studies on the erastin-resistant clones that we would like to address. Regarding whether the resistant cell lines are still able to undergo ferroptosis in response to different cell death inducers, we note that in Figure 8, we showed that these resistant cell lines were resistant not only to erastin, but also to sulfasalazine and sorafenib, two additional validated ferroptosis-inducers ([13] Cell and this report).

To investigate this question further, we analyzed the sensitivity of the parental cell line (DU-145) and three of the resistant clones (2.5E-1, 2.5E-3 and 2.6E-23) to several of the more potent erastin analogs described in Figure 3, as well as to an additional structurally- and functionally-unrelated inducer of ferroptosis, (1S,3*R*)-RSL3, which triggers ferroptosis by inhibiting glutathione peroxidase 4 (GPX4) ([66] Cell). We found that the three resistant cell lines exhibit resistance to all tested ferroptosis inducers, including (1S-3*R*)-RSL3. These results are shown below and are now included as Figure 8—figure supplement 1. These data further supports our model that these cell lines are resistant to a broad spectrum of ferroptosis inducers, consistent with the resistance arising downstream in pathway, i.e., after lipid ROS generation.

Regarding the second point, in the resistant clones, we see high levels of expression of multiple *AKR1C* family members, as well as other antioxidant genes that may play a role in the resistance of these clones to ferroptosis. We suspect that it is unlikely that knock down of any single one of these genes will resensitize cells to erastin, as they likely have redundant function. Nonetheless, we attempted to re-sensitize erastin-resistant DU-145 clones by modulating *AKR1C1* levels through the use of small interfering RNAs (siRNAs). We observed, as expected, that silencing of *AKR1C1* alone was insufficient to confer substantial re-sensitization (not shown), likely because the multiple *AKR1C* family members upregulated simultaneously have redundant function. Further analysis of the overlapping role of these family members will require substantially more work and we believe that this is beyond the scope of the current manuscript.

*7) In previous work (Yagoda 2007), the authors showed that erastin bound to SLC7A5, a subunit of system L, and they mention in Dixon 2012 that treatment with erastin resulted in highly significant decreases in the levels of system L substrates. These findings appear to contrast to the observations in the current work that erastin is selective for system x- and did not affect system L. The authors should discuss these apparently contradictory results in the text*.

AWe thank the reviewer for raising this point so that we can clarify this issue. We now address this point in the first paragraph of the Discussion section of the updated text, where we clarify our current understanding of the erastin mechanism of action as follows:

“Previous metabolomics analysis had suggested that erastin inhibited system L (SLC3A2 + SLC7A5)-mediated amino acid transport in Jurkat T cells (13). It was therefore surprising to observe that in HT-1080 and Calu-1 cells erastin inhibited system x_c_^-^ (SLC3A2 + SLC7A11)-mediated cystine uptake but not system L-mediated phenylalanine uptake. These results rule out the possibility that erastin inhibits SLC3A2-dependent transporters non-specifically. Further, given evidence that Jurkat cells do not express system x_c_^-^ (31), we hypothesize that in cells lacking system x_c_^-^, erastin can inhibit structurally-related transporters (e.g., system L). An alternative hypothesis is that erastin binds to some indirect target that, in Jurkat cells, favors the inhibition of system L, while in HT-1080, Calu-1 and possibly other cells, favors the inhibition of system x_c_^-^. A definitive resolution of this matter will require further study.”

*8) The authors made 19 analogs including several with significantly improved potency. They do not show the selectivity of these compounds for RAS mutant cells, which would be important for therapeutic use. A previous study from this group did report selectivity for a smaller set of analogs (Yang 2014). The selectivity should be investigated for the best analogs*.

In the updated version of the manuscript, we report the selectivity of the most potent analog (21), the parent compound (3) and an inactive analog (14) in the BJ cell system. Consistent with previous results and with our expectations, we find that 21 and 3 exhibit greater potency in BJeLR (BJ fibroblast + human telomerase + small T antigen + large T antigen + oncogenic HRAS^V12^) versus BJeH (BJ fibroblast + human telomerase) and that the improved analog 21 is substantially more potent, overall, than 3. These new data are shown below and in updated Figure 3. In the accompanying text we now say:

“Previously we have shown that erastin and lethal analogs thereof demonstrate selective lethality towards human BJ fibroblasts engineered to express human telomerase, SV40 large and small T antigen, and oncogenic HRAS^V12^ (BJeLR) compared to isogenic cells expressing only telomerase (BJeH) (16, 66). We tested the most potent lethal analog (21), along with the parent compound (3) and a representative non-lethal analog (14), in these cell lines. While 14 was inactive, we found that both lethal analogs (21 and 3) retained selectivity towards BJeLR versus BJeH cells (Figure 3). Consistent with the pattern of lethality observed in HT-1080 cells, 21 was a more than 20-fold more potent lethal molecule compared to 3 (BJeLR EC_50_ of 22 nM [95% C.I. 20-25 nM] versus 490 nM [95% C.I. 350-690 nM], respectively).“

*9) A statistical analysis (*Figure 7*) reveals sorafenib to cause multiple side effects in patients that might be partially linked to System x*_*c*_^*-*^
*inhibition beyond its well-documented kinase inhibitory effects. As these are mere statistical data, the statement that sorafenib indeed causes these side effects in patients partially due to system x*_*c*_^*-*^
*inhibition is over-interpreted and should be de-emphasized. In that sense, the authors should consider that mice lacking system x*_*c*_^*-*^
*are fully viable and only show subtle phenotypes (Sato et al., JBC 2005). Therefore, major adverse effects induced by system x*_*c*_^*-*^
*inhibition in “normal” cells are less likely, which questions the validity of the statement on page 22, second paragraph, last sentence. Furthermore, the paper by*
[11]
*(referred to on page 27) describes that xCT knockout cells are in fact more resistant to against limbic seizures, and 6-OH dopamine lesioning in substantia nigra (Massie et al., FASEB 2011)*.

Regarding the potential connection between sorafenib, system x_c_^-^ inhibition and adverse effects observed in patient records, we agree with the reviewer concerning the correlative nature of these data. All drugs have idiosyncratic side effects in a small number of patients—some of these are specific to the drug and some are specific to the mechanism of action of the drug. We speculate here that the small percentage of patients that exhibit side effects is linked to the lethal mechanism of sorafenib, and specifically that a kinase inhibitor that also inhibits system x_c_^-^ would cause a different spectrum of idiosyncratic side effects compared to a kinase inhibitor that does not cause system x_c_^-^ inhibition. Previously we demonstrated that none of the other comparison kinase inhibitors triggered ferroptosis. We have collected new data, shown below and incorporated as Figure 7—figure supplement 1, which demonstrates that none of these comparison kinase inhibitors inhibit glutamate release. In the revised text we now state:

“These results suggest that, compared to other clinically-approved kinase inhibitors, sorafenib treatment has an increased propensity generate unexpected adverse events in a variety of physiological systems. While these data are merely correlative at this point, one possibility, given the unique ability of sorafenib to inhibit system x_c_^-^ among tested kinase inhibitors, is that this effect contributes to increase the chance of an adverse event in combination with one or more underlying modifying factors.”

We believe that this statement is consistent with our results and suggests a viable hypothesis to explain the observed correlation. Further work will be required to investigate this correlation, but that lies beyond the scope of this report. In the updated Discussion we also remove the reference to the work of De Bundel (2011) as we agree that this could be confusing.

We would also like comment on the reviewers’ point concerning the subtle effects observed in mice lacking the system x_c_^-^ component *Slc7a11* (e.g., [46] J Biol Chem; [7] PNAS). One interpretation of these data is that no drug targeting system x_c_^-^ in vivo would be expected to result in any pronounced physiological effect. Certainly the effects of many drugs are effectively modeled by the corresponding deletion allele in mouse (discussed in Zambrowicz & Sands, 2003 Nat Rev Drug Discovery). However, this need not always be true, especially for drugs that target metabolic networks. This is because metabolic networks frequently contain redundant synthetic routes to the same product (e.g., Kitami & Nadeau, 2002 Nat Genet). Thus, inactivation of a single node can be buffered by the use of alternative pathways to the same end product. In fact, it is clear from our RNA Seq analysis that in response to erastin treatment (→ system x_c_^-^ blockade) cells upregulate a key enzyme in the transsulferation pathway (CBS), which provides an alternative route to the synthesis of cysteine from methionine. While speculative, in the *Slc7a11* knockout mouse it is likely that compensatory genes (such as Cbs) and pathways (such as transsulfuration) are gradually activated in cells lacking system x_c_^-^ function, enabling cell survival. Conversely, acute drug treatment may not allow sufficient time for compensatory upregulation of alternative pathways to occur, resulting in cell death.